# LLPS REDIFINE allows the biophysical characterization of multicomponent condensates without tags or labels

Mihajlo Novakovic [1] ✉, Yaning Han[1,2], Nina C. Kathe [1,2], Yinan Ni [1,2], Leonidas Emmanouilidis [1] ✉ & Frédéric H.-T. Allain [1] ✉

Liquid-liquid phase separation (LLPS) phenomenon plays a vital role in multiple cell biology processes, providing a mechanism to concentrate biomolecules and promote cellular reactions locally. Despite its significance in biology, there is a lack of conventional techniques suitable for studying biphasic samples in their biologically relevant form. Here, we present a label-free and non-invasive approach to characterize biomolecular condensates termed LLPS REstricted DIFusion of INvisible speciEs (REDIFINE). Relying on diffusion NMR measurements, REDIFINE exploits the exchange dynamics between molecules in the condensed and dispersed phases to determine not only diffusion constants and the fractions in both phases but also the average radius of the condensed droplets and the exchange rate between the phases. Observing proteins, RNAs, water, as well as small molecules, and even assessing the concentrations of biomolecules in both phases, REDIFINE analysis allows a rapid biophysical characterization of multicomponent condensates which is important to understand their functional roles. In comparing multiple systems, REDIFINE reveals that folded RNA-binding proteins form smaller and more dynamic droplets compared to the disordered ones.

Cells organize complex biochemical reactions in space and time by forming compartments[1-3]. Besides the organelles enclosed by membranes, proteins and nucleic acids can phase separate into membraneless liquid-like compartments such as the prominent nucleolus, nuclear speckles, processing bodies, or stress granules[4-6]. These cellular structures, formed through the process of liquid-liquid phase separation (LLPS), have been described as coacervates[7,8] and are optically resolvable as micron-sized structures. Given their involvement in RNA metabolism, many of these biomolecular condensates are enriched in proteins that contain RNA-binding domains (RBDs) and LLPS-prone intrinsically disordered regions (IDRs)[9-11].

Numerous RNA-binding proteins (RBPs) undergo LLPS under physiological conditions, either in the presence or absence of RNA. One of the examples of RBPs that recently attracted a lot of attention in the scientific community during the COVID-19 pandemic, SARS-CoV-2

Nucleocapsid (Nu), binds the RNA and engages in multivalent homo- and heterotypic interactions leading to LLPS[12,13]. It is hypothesized that its phase separation with RNA is essential for genome RNA packaging and the formation of viral ribonucleoprotein (vRNP)[14,15]. Another RBP protein, Polypyrimidine tract-binding protein 1 (PTBP1), can form condensates with RNA, driving the production of the compartments in the nucleus to promote gene silencing[16,17]. Moreover, LLPS is associated with several diseases, including amyotrophic lateral sclerosis (ALS) and frontotemporal dementia, where stress may induce yet another RNA-binding protein FUS (FUsed in Sarcoma)[18,19] to form granules that mature into solid aggregates. Disease-associated mutations accelerate this liquid-to-solid transition[20]. Besides FUS, a mammalian DEAD-box helicase (DDX4) can also undergo LLPS by itself (by self-association of the disordered N terminus)[21], forming condensed droplets that can act as a molecular filter concentrating single-

[1]Department of Biology, Institute of Biochemistry, ETH Zurich, Zurich, Switzerland. [2]These authors contributed equally: Yaning Han, Nina C. Kathe, Yinan Ni. ✉e-mail: mihajlo.novakovic@bc.biol.ethz.ch; leonidas@bc.biol.ethz.ch; allain@bc.biol.ethz.ch

stranded DNA but excluding double-stranded DNA[22]. The domain schemes of these proteins are shown in Supplementary Fig. 1.

Despite the evidence of the functional importance of phase-separation for these proteins, a wide spectrum of questions about the biophysical properties of these biomolecular condensates and their relation to (patho)physiology in biological systems remain unanswered[23,24], due to a lack of suitable experimental procedures that are not invasive and that can probe the biomolecules in their relevant form[25–27]. Although multicolor labeling can provide a plethora of information about the condensates, the fluorescent tags often affect protein conformation and dynamics, which can influence their phase separation, introducing potential bias in the conclusions[28–34].

In this work, we examined several biomolecular condensates (protein and protein-RNA complexes) using LLPS REDIFINE, an approach that utilizes the restricted diffusion of biomolecules inside the droplets and chemical exchange as an NMR lens to probe the physical properties of the condensates. Without the need for specific labels, we could comprehensively characterize the biomolecular condensates formed by wild-type FUS, DDX4, Nu, and PTBP1 proteins and correlate the extracted information to their biophysical characteristics.

## Results

### REDIFINE provides diffusion constants, exchange rates, partitioning and droplet size of a biphasic protein LLPS system

For our first application of LLPS REDIFINE, we examined the biphasic sample of the N-terminal domain (NTD) of the human FUS protein, which we studied earlier[19]. It comprises the low complexity QGSY-rich segment and the first arginine-glycine repeat (RGG1)[19]. FUS NTD domain is completely unstructured (Supplementary Fig. 1) and is considered to be essential for the phase separation of FUS protein[35]. FUS NTD droplets were stabilized in vitro using an agarose hydrogel, which allows prolonged spectroscopical analysis (Supplementary Fig. 2)[19]. Previously, we have shown that the large diffusion difference between FUS NTD in dilute and condensed phases can be used to quantify their respective populations[19]. However, the choice of longer diffusion times, $\Delta$, could impair this quantification due to the presence of protein exchange between the two phases. With this in mind, we performed multiple diffusion measurements on FUS NTD biphasic sample stabilized in agarose, where we varied the gradient strength as well as the diffusion time $\Delta$ (Fig. 1a). As expected, the decay of protein signal is more pronounced at the longer diffusion times. Figure 1b

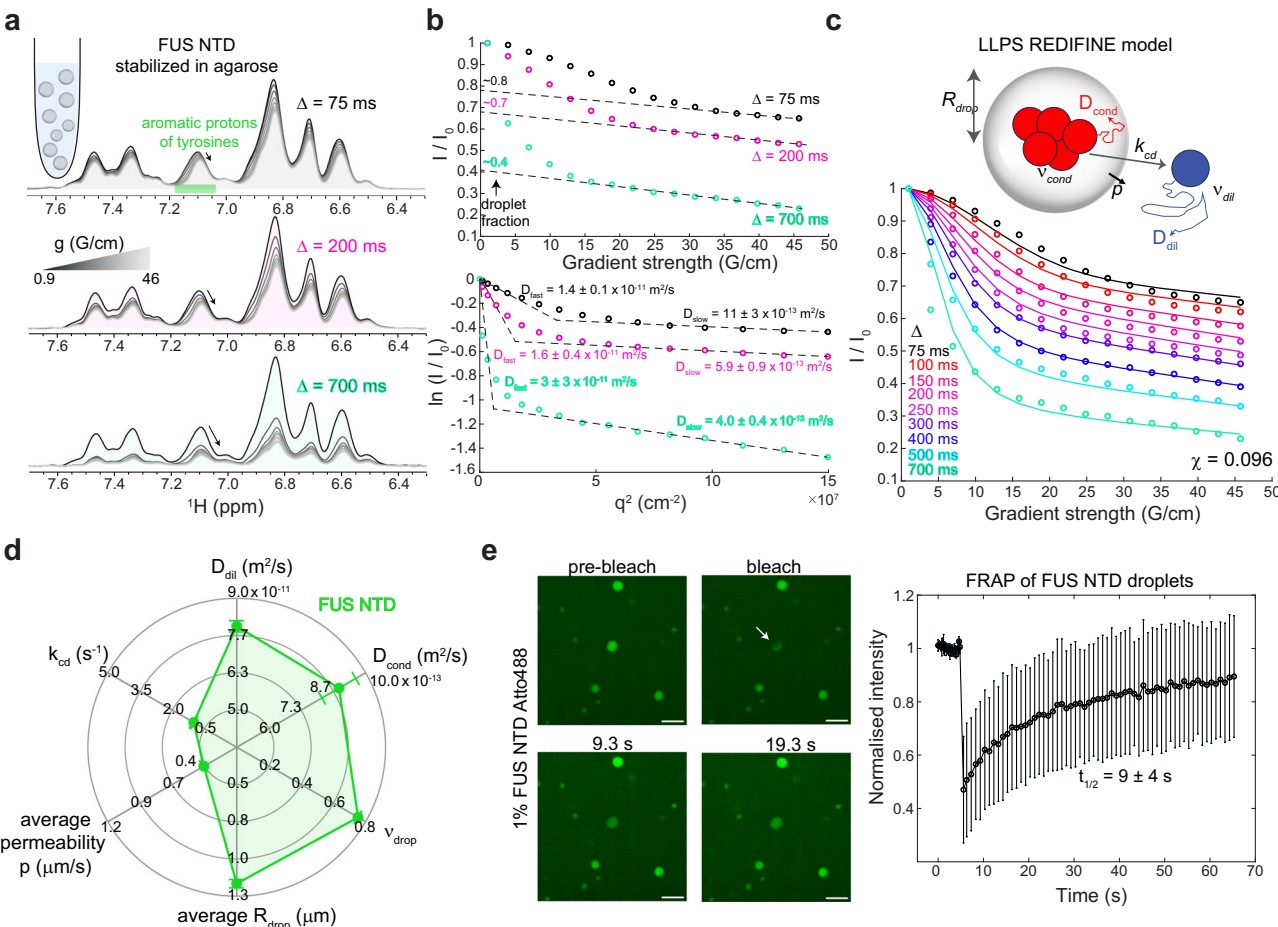

**Fig. 1 | FUS NTD characterization. a** A series of 1D proton spectra showing signal decay with increasing gradient strength in diffusion NMR experiment acquired on fresh FUS NTD sample using different diffusion times. Resonance highlighted in green contains an overlapping signals of tyrosine aromatic protons from both protein in dilute and condensed phase dispersed in agarose. **b** Decay of the integral of tyrosine signals plotted as $I/I_0 = f(g)$ and $\ln(I/I_0) = f(q^2)$ illustrating the influence of diffusion time on droplet fraction and diffusion coefficients. **c** LLPS REDIFINE model and the ensuing fitting of the FUS NTD diffusion data set. Following parameters are determined: $D_{dil} = 8.0 \pm 0.2 \times 10^{-11}$ and $D_{cond} = 8.9 \pm 0.7 \times 10^{-13}$ m²/s, $v_{cond} = 0.753 \pm 0.002$, $R_{drop} = 1.21 \pm 0.03$ µm, $p = 0.405 \pm 0.007$ µm/s and

$k_{cd} = 1.00 \pm 0.03$ s⁻¹ is derived from $R_{drop}$ and $p$. **d** Spider chart summarizes the properties of FUS NTD condensate and represents its REDIFINE fingerprint. 10 ms gradient length was used in (**a**, **b**). The final value for minimization function $\chi$ indicating the goodness of the fit is shown in (**c**). The uncertainties for each parameter are calculated using covariance matrix and report on the ambiguity of determined values. **e** FRAP experiment performed on FUS NTD spiked with 1% Atto488-labeled FUS NTD in 0.5% agarose. Scale bar is 5 µm. Data are presented as mean values +/− the standard deviation from the measurements on 4 different droplets. Source data are provided as a Source Data file.

illustrates that indeed, the population of the condensed phase that would be extrapolated from the plateau[19] becomes lower with increasing $\Delta$. Furthermore, when the signal decays are plotted as the logarithmic value of normalized signal with respect to the global diffusion parameter $q^2$ as defined in the theoretical section of the Methods, the diffusion constants can be calculated directly from the slope of the resulting lines (Supplementary Note 1). We observed that the apparent diffusion constants of both the slow- and fast-diffusing proteins are also strongly influenced by the diffusion time $\Delta$ (Fig. 1b). This implies that these experimental decay curves cannot be simply explained by two independent protein populations and that only by taking into account exchange between the populations one could interpret these curves.

We therefore introduced the model outlined in the theoretical section of the Methods and in Notes 2 and 3 of Supplementary Information to interpret these decay curves. Our model considers a population of protein $\nu_{cond}$ inside the condensed droplets and $\nu_{dil}$ in the dilute phase in equilibrium that can exchange between each other, given by $\nu_{cond}k_{cd} = \nu_{dil}k_{dc}$ where $k_{cd}$ and $k_{dc}$ are the forward and reverse exchange rates, respectively. The average size of the droplets is described by the radius $R_{drop}$ and the protein inside experiences a restricted diffusion with the apparent diffusion constant $D_{cond}^{app}$ that is dependent on the droplet size, the gradient strength g, and diffusion time $\Delta$ (compared to the isotropic $D_{dil}$ in the pure dilute phase)[36,37]. The droplet-bulk interface is characterized by a permeability for a molecule to move between the phases, and it is defined as the permeability factor $p$. The exchange rate from condensed to dilute phase $k_{cd}$ is expressed as $k_{cd} = \frac{3p}{R_{drop}}$ where $\frac{3}{R_{drop}}$ the surface-to-volume ratio of a sphere[38]. As the $D_{cond}^{app}$ is too slow to be measured by conventional NMR probes due to technical limitations, our method exploits the chemical exchange between the two phases to relay the information about slow restricted diffusion onto more easily observable NMR signals of the dilute phase. Hence the name LLPS REstricted DIFusion of INvisible speciEs (REDIFINE).

The framework of REDIFINE model is illustrated in Supplementary Note 2. We ran extensive simulations to show how various parameters including $R_{drop}$, permeability and $\nu_{cond}$ affect diffusion curves. Not surprisingly, the choice of diffusion time $\Delta$ influences the extent of the modulation in diffusion curves caused by different condensate parameters (Supplementary Note 2). Using REDIFINE simulation, we could fully reproduce the acquired data for FUS NTD. As the chemical exchange is not a priori known, our simulations also revealed that in order to unambiguously determine condensate parameters by REDIFINE, multiple experiments with several diffusion times $\Delta$, are required (Supplementary Note 2). To test this, we subjected FUS NTD sample prepared in agarose to LLPS REDIFINE, integrating signals stemming from the aromatic protons of tyrosines (Fig. 1a), which were assigned based on their characteristic chemical shift. REDIFINE provided excellent global fitting to the curves (Fig. 1c). In just 2 h of experimental recording followed by REDIFINE fitting, key FUS LLPS properties could be determined including FUS diffusion constants in both phases ($8.0 \times 10^{-11}$ and $8.9 \times 10^{-13}$ m²/s for the dilute and the condensed phases, respectively), the fraction of the protein in the droplets ($\nu_{cond}$ of 0.75 at 600 µM FUS NTD) and the droplets average radius (1.2 µm) as well as their permeability (0.4 µm/s) from which we can derive the exchange rate from condensed to dilute phase ($k_{cd} = 1$ s⁻¹). These parameters of the biphasic sample obtained with REDIFINE can be illustrated by the spider chart shown in Fig. 1d, which represents a fingerprint for FUS NTD LLPS. We can also define a global exchange rate that accounts for the partitioning between phases $k_{ex} = \nu_{cond}k_{cd} = \nu_{dil}k_{dc}$, which can be somewhat comparable to the rates obtained in fluorescence recovery after photobleaching (FRAP)[39,40] experiments. This results in a $k_{ex}$ rate of

0.75 s⁻¹, corresponding to a half-life time ($t_{1/2}$) of 0.9 s according to the equation $t_{1/2} = \frac{\ln 2}{k_{ex}}$. This result is very consistent with the FRAP data acquired on the FUS QGSY-rich region[41] (recovery $t_{1/2} \sim 1$s) and yet with another unstructured protein, α-Synuclein[42] (recovery $t_{1/2} \sim 3$s). LLPS REDIFINE droplet size is also in very good agreement with the values determined by a thorough analysis of fluorescent images taken for FUS NTD in agarose[19], which further indicated that FUS NTD droplet size does not depend on initial protein concentration. We prepared two additional FUS NTD samples at a concentration of 200 µM and measured REDIFINE three times for each sample. Supplementary Note 4 summarizes these measurements and confirms very consistent fitting parameters throughout the technical and biological replicates.

As an independent control, we prepared a fresh sample of FUS NTD in 0.5% agarose and visualized it under the microscope (Supplementary Fig. 2a). FUS NTD droplets stabilized in agarose indeed have a very uniform size of circa 1 µm in radius, validating the REDIFINE methodology. To test droplet dynamics, we performed FRAP experiments by adding 1% FUS NTD which was labeled with Atto488 dye (Supplementary Fig. 2b). We chose the small dye Atto488 over a fusion protein to minimize the effect of the labeling on the phase separation of FUS NTD. Figure 1e shows the fluorescence recovery upon bleaching of the FUS NTD droplet, indicating liquid-like condensate properties. Due to the low bleaching efficiency of Atto488 dye, the full laser power was required to reach a significant reduction in fluorescence. The FRAP recovery that we obtained ($9 \pm 4$ s⁻¹) was slower than the REDIFINE exchange rate and the literature values. We attribute this discrepancy to the local heating effects caused by the high-power irradiation, as the bleached droplet slightly changed shape over time (Fig. 1e). This could also be a reason for the lower recovery levels that we achieved, as high irradiation power can cause degradation and/or enhanced oligomerization of FUS NTD.

## LLPS REDIFINE applied to full-length FUS and water

Next, we extended REDIFINE to the full-length FUS, which, despite the presence of two folded RNA binding domains, is still mostly unstructured (with only 8.9 % of the sequence adopting a stable structure). Akin to FUS NTD, full-length FUS spontaneously forms droplets by itself. At a concentration of 200 µM, the large majority of full-length FUS is found inside the condensed phase based on an almost non-decaying signal in NMR diffusion experiment[19] (Fig. 2a). Supplementary Fig. 3a shows the LLPS REDIFINE fitting for this FUS biphasic sample. From REDIFINE, we indeed found that 90% of the protein is inside the droplets ($\nu_{cond} = 0.9$, translating to 1.6 mg of protein) and consequently very low population of FUS outside which is primarily why the diffusion constant of the protein in the dilute phase could not be accurately determined. Furthermore, in this biphasic condensate of FUS full-length, we could quite surprisingly spectroscopically observe a small population of slowly diffusing water with a shifted frequency implying different interactions in the condensed phase. This population corresponds to structured water[43–45] found in the droplets, and as the water decay curve showed a complex biexponential shape (Fig. 2b), we could apply LLPS REDIFINE to the water signal as well. Supplementary Fig. 3b presents an excellent fit of the water LLPS REDIFINE data, confirming a droplet radius of around 1 µm consistent with the radius obtained independently using the FUS FL signal decay. REDIFINE revealed also that there is around 3% of water trapped in the droplets (accounting for a volume of 4 µl of water). Similar results are detected for FUS NTD, though with slightly lower water content in the droplets (Supplementary Note 5). Due to such a small population of water in the droplets, the uncertainty for the diffusion coefficient in the condensed phase is large, but still revealed that water is diffusing two to three orders of magnitude slower than in the dispersed phase

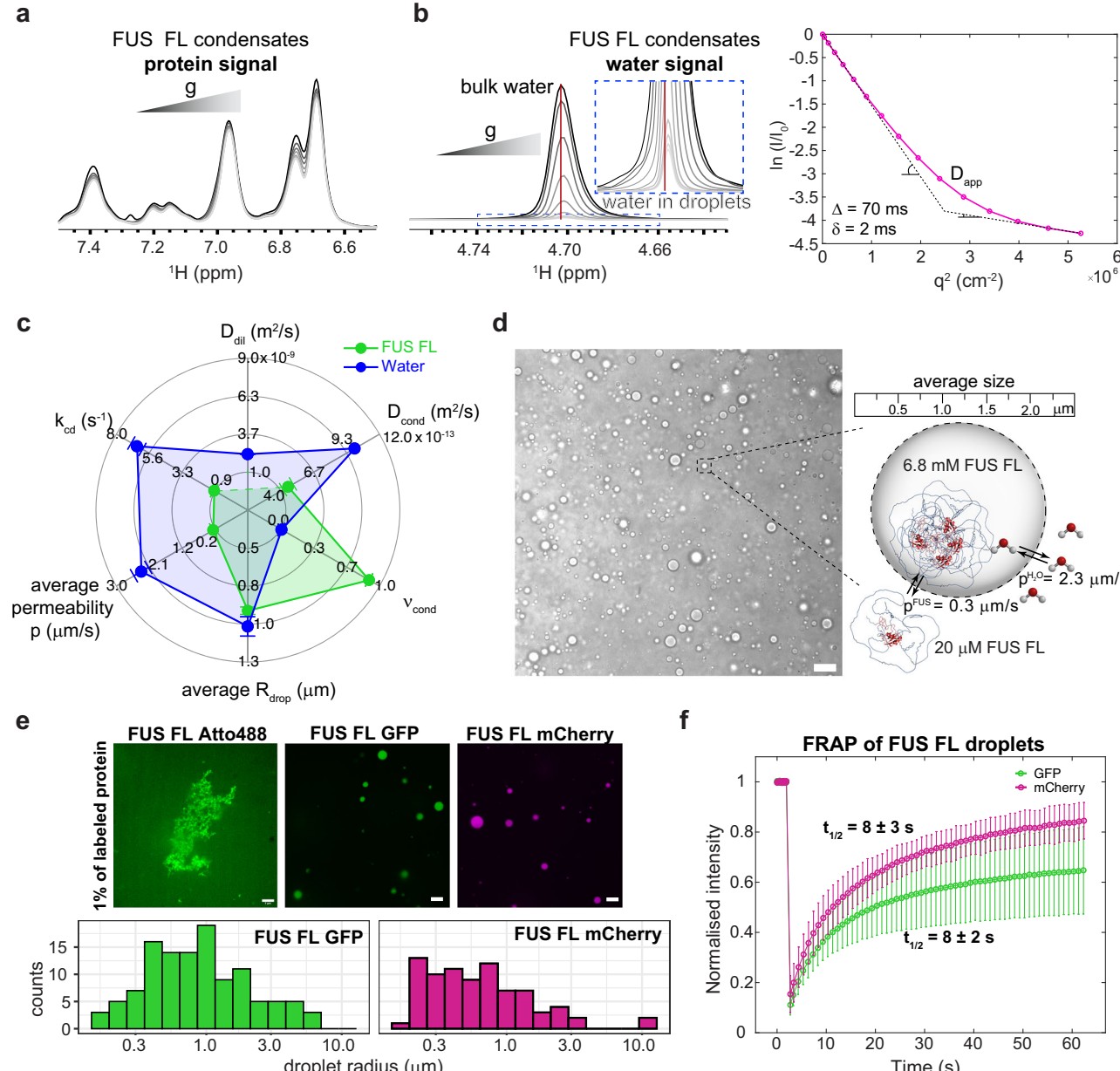

**Fig. 2 | Application of LLPS REDIFINE to full-length FUS condensates. a** A series of 1D proton spectra showing full-length FUS signal with increasing gradient strength in NMR diffusion experiment. Non-decaying signal implies a very high population of slow-diffusing FUS in the droplet. **b** Similarly, a series of 1D proton spectra showing the water signal from the same FUS sample. Note that based on different chemical shifts, there is a distinct population of bound water in the droplets. Water diffusion also exhibits biexponential behavior. **c** Summary of LLPS REDIFINE data set acquired on both protein and water signal, providing multi-component characterization of FUS FL condensate. **d** LLPS REDIFINE allows us to "visualize" the FUS FL condensates, providing average droplet size, interface permeabilities, and concentrations in corresponding phases. Scale bar is 10 μm.

**e** Fluorescent images acquired on FUS FL sample using different fluorescent tags. Analysis of droplet sizes could be performed in the case of GFP and mCherry tags, while the addition of 1% of Atto488-labeled protein causes aggregation. Scale bar is 5 μm. **f** FRAP recovery time course for GFP and mCherry tags. Data are presented as mean values +/− the standard deviation from the measurements on 4 different droplets. Experiments in (**a**) are acquired using 75 ms diffusion encoding time and 10 ms gradient length while in (**b**) for water 70 ms and 2 ms respectively. The uncertainties for each parameter in (**c**) are calculated using a covariance matrix and report on the ambiguity of determined values. The representative images are from three independent experiments. Source data are provided as a Source Data file.

and within the same range as the FUS protein. FUS condensate parameters of the protein and the water are summarized in Fig. 2c, d. As expected, they both report the same size, however the interface permeability for water is 3-fold larger than for the protein. Knowing the initial amounts of protein and water, and accounting for protein and water fractions in the condensed phase result in a FUS concentration of 6.8 mM in the droplets and correspondingly 20 μM in the dilute phase (340−fold less) as illustrated in Fig. 2d. Although these correspond to the concentrations of a macromolecule in a biphasic LLPS

sample in equilibrium, our result is in agreement with previously reported concentrations for the FUS protein in a single condensed phase after physical separation of the two phases[41].

Again, to independently compare our measurements with other methods, we added 1% of Atto488 labeled protein to visualize condensates by fluorescent microscopy. This labeling unfortunately caused the whole sample of FUS FL protein to aggregate (Fig. 2e) possibly due to the presence of 15 lysine residues in FL protein where the dye can be potentially bound affecting the net charge and leading

to cumulative effects of increased hydrophobic interactions. On the contrary, by spiking FUS FL with 1% GFP- and mCherry-tagged fusion protein, we could observe liquid-like FUS FL droplets. The size distribution analysis of the fluorescent images showed that the droplet radius average was very slightly below 1 μm with the mCherry droplets being smaller than the GFP-tagged ones. FRAP experiment showed similar recovery for both GFP- and mCherry-tagged protein (Fig. 2f) while the aggregated sample with Atto488 label never recovered. Akin to the FUS NTD, the FRAP recovery rates are by the same amount slower compared to REDIFINE chemical exchange. Impaired bleaching recovery correlated with an observation that FRAP rates closely depend on the size of the droplet that was bleached (Supplementary Fig. 4a) might be explained by the disproportionate partitioning of the tagged protein between the phases[46]. Faster FRAP recovery detected after partial bleaching of larger droplets compared to the small ones can be rationalized by the steep reduction in exchange efficiency when the fraction of protein in the dilute phase is below 10% (Supplementary Fig. 4a, b). As the exchange process between phases depends on droplet size and populations in each phase, REDIFINE provides much more comprehensive analysis of the condensates than FRAP. In addition, the example of FUS FL further shows evidence that even a small amount of fluorescent tag can affect phase separation and even introduce bias in the fluorescent measurement raising the demand for the development of label-free methodologies.

## Independent validation of exchange rates by FEXSY

Still puzzled by the difference between REDIFINE and FRAP, we analyzed another intrinsically disordered N-terminal domain from DDX4 protein ($SSP = 0.052$, Supplementary Figs. 1 and 5a). Diffusion constant in the condensed phase of around $7 \times 10^{-13}$ m²/s that we determined with REDIFINE perfectly matched with the value obtained on a monophasic dense phase of DDX4 using very elegant NMR experiments that exploit methyl triple-quantum states for diffusion encoding[47]. Besides the protein-specific diffusion and population parameters, we could observe that the average droplet size ($R_{drop} = 1.05$ μm) and global chemical exchange rate ($k_{ex} = 0.61$ s⁻¹) of DDX4 condensates is very similar to FUS indicating a potentially common mechanism of condensate formation for unstructured proteins. On this sample, we also measured an independent NMR experiment that could be adapted to LLPS systems to measure directly the rate of chemical exchange between phases. Filter-Exchange NMR Spectroscopy (FEXSY)[48–50] can filter out the signal stemming from the dilute phase and consequently measure the rate of signal recovery as a result of chemical exchange from the condensed phase. The fitting of FEXSY datasets confirmed the same

exchange rate and protein fraction in the condensed phase as determined by LLPS REDIFINE (Supplementary Fig. 5b). To further test this, we compared REDIFINE and FEXSY on another biphasic sample of FUS FL incubated for two months at room temperature (Supplementary Fig. 5c, d). Again, the chemical exchange determined by REDIFINE and FEXSY closely matched. Note that due to its lower sensitivity, FEXSY experiments took 5-fold longer to acquire compared to LLPS REDIFINE but could not report on interface permeabilities, droplet size, and reliably on diffusion coefficients.

## REDIFINE reveals the biophysical properties of bimolecular RBP-RNA condensates

Besides previously discussed systems, many proteins can undergo LLPS only in the presence of binding molecules, forming bi- or multi-molecular condensates[3]. A well-known example is the SARS-CoV-2 Nucleocapsid protein that binds negatively charged RNA, which in turn induces condensate formation. Nu readily phase-separates with sub-stoichiometric quantities of structured viral RNA fragments such as the stem-loop-2 s2m RNA (Supplementary Fig. 6a, b)[51]. DOSY decay curves acquired on free Nu can be easily fitted to a single exponential decay indicative of a single dispersed phase (Supplementary Fig. 7a). With RNA present, phase separation occurs, and decay curves experience a steep decrease of the signal with no significant plateau compared to FUS samples. The data could not anymore be quantitatively explained by a single exponential function, however fitting with LLPS REDIFINE (Supplementary Fig. 7b, c), allowed delineating the exchange-averaged diffusion information about the Nu condensates. $D_{cond}$ is circa two orders of magnitude slower than $D_{dil}$, and around 27% of the protein resides in the droplets. The average exchange rate is determined to be 4.8 s⁻¹ which is faster than for fully unstructured proteins, explaining the steeper diffusion decay at higher gradient strengths. The size of the droplets determined by REDIFINE is around 0.8 μm, which is confirmed by fluorescence microscopy (Supplementary Fig. 7d, e). Yet, using this complex, we could not detect the RNA and therefore could not probe the exchange and the diffusion of the RNA in the condensed phase.

As Nucleocapsid protein can promiscuously interact and form droplets even with short single-stranded RNAs, we then examined a Nu condensate bound to the 6-mer poly-A RNA (A6). Maximum turbidity is reached at 6 equivalents of this single-stranded RNA (Supplementary Figs. 6a, b and 8a). Unlike s2m, the NMR RNA signals are now detectable allowing us to apply LLPS REDIFINE to both the protein and RNA signals (Fig. 3a). This is equivalent to multicolor fluorescent labeling, however without any specific dye as every type of polymer possesses a distinct the NMR signature. Figure 3a illustrates the aliphatic peaks of

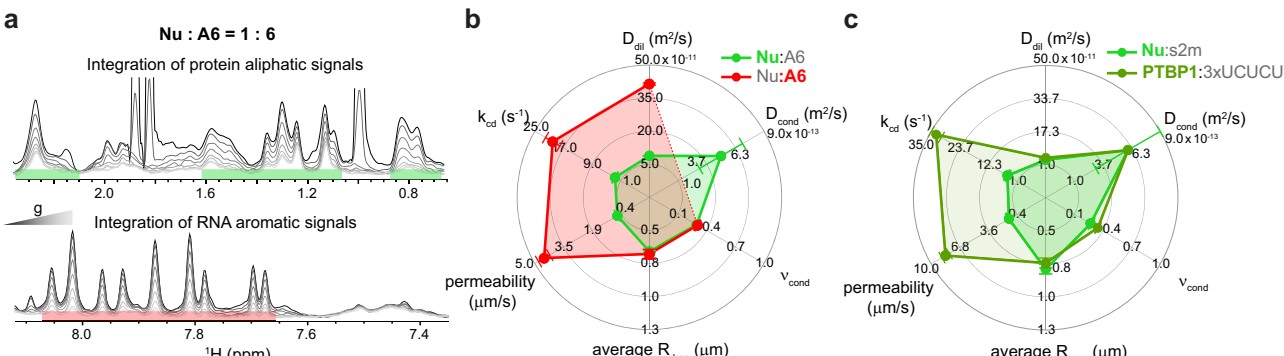

**Fig. 3 | Application of LLPS REDIFINE to structured proteins that phase separate upon addition of RNA. a** Nucleocapsid:A6 condensates prepared at 1:6 protein:RNA ratio. Both protein and RNA peaks are visible in NMR spectrum and are analyzed separately by LLPS REDIFINE. Spectra are acquired using 75 ms diffusion time and a gradient length of 10 ms for protein and 3.5 ms for RNA. **b** LLPS REDIFINE results obtained from both Nucleocapsid and RNA side. **c** Overlay of condensate

properties obtained on condensates formed by Nucleocapsid protein and structured s2m RNA and by another structured protein, PTBP1 in the presence of 3 × UCUCU RNA. The uncertainties for each parameter are calculated using covariance matrix and report on the ambiguity of determined values. Source data are provided as a Source Data file.

the protein and aromatic resonances of the A6 RNA that we used for observing the decay and fitting with REDIFINE. Reassuringly, both protein and RNA (Supplementary Fig. 8b and Fig. 3b) data sets independently report the same average droplet size but more surprisingly we obtained a significantly different permeability for the interphase exchange. We could see that A6 RNA exchanges almost eight-fold faster than the Nucleocapsid protein, probably due to being smaller and in large excess. Although having a better fit ($\chi = 0.23$ vs 0.70), the uncertainty of the RNA's $D_{cond}$ is large (Supplementary Fig. 8b) due to fast chemical exchange averaging. From the equal fractions of protein and RNA in the droplet (-0.3), it can be implied that the condensate contains 6 equivalents of A6 for one Nu protein in both phases, which is the ratio needed to reach maximum turbidity (Supplementary Fig. 6b).

In contrast to the Nucleocapsid that promiscuously binds the RNA, we also investigated the multidomain PTBP1 protein that forms a specific complex of high affinity with an RNA containing several UCUCU motifs. With 0.1 equivalent of 3 × UCUCU RNA (Supplementary Figs. 6a, b, and 8c), PTBP1 readily phase separates, and at room temperature, around 50% of PTBP1 is found in droplets (consistent with independent spectrophotometric measurements). Surprisingly, we found that PTBP1 is exchanging very quickly between the condensed and dilute phases (90 s$^{-1}$, see Supplementary Fig. 8d). These rates are very high on the diffusion time-scale (tens to hundreds of milliseconds) and REDIFINE could not determine unambiguously the diffusion coefficients and the droplet size. However, at 278 K the droplets are still present in the sample and the $k_{cd}$ rate is significantly reduced (around three-fold) allowing REDIFINE to provide information about

the droplet size and permeability as well as the diffusion of PTBP1 in the condensed phase (Supplementary Fig. 8e, and Fig. 3c). The observed reduction in the exchange is consistent with the Arrhenius temperature dependence of kinetic rates. For comparison, Nu:s2m results from Supplementary Fig. 7 are also presented in Fig. 3c, showing a very similar REDIFINE fingerprint to Nu:A6 condensate and even to PTBP1:3 × UCUCU in terms of diffusion constant, size, and fraction of proteins, despite being different in terms of droplet permeability and exchange. This suggests that RNA-induced condensates have similar droplet properties. Altogether, we showed here that LLPS REDIFINE allows the biophysical characterization of multicomponent biological condensates such as protein-RNA condensates (Supplementary Table 1), and the method could be easily extended to multiprotein or drug-protein condensates.

## Protein structure and RNA modulate the condensate properties

After examining more than 15 different condensate samples involving several different proteins and RNAs (Supplementary Table 1), we indeed observed an intriguing trend in the droplet properties among different systems. The condensates of unstructured proteins generally tend to form larger droplets and experience slower interphase exchange (i.e., FUS, DDX4). Proteins with structured domains that require the addition of RNA to phase separate exchange faster and usually form smaller droplets (i.e., Nu, PTBP1). This is illustrated in Fig. 4a by a 2D plot showing an inverse correlation between the condensed-to-dilute exchange rate and the droplet size. Interestingly, the FUS biphasic sample in the presence of RNA (green dot) contains droplets that are more dynamic and closer to the region defined by

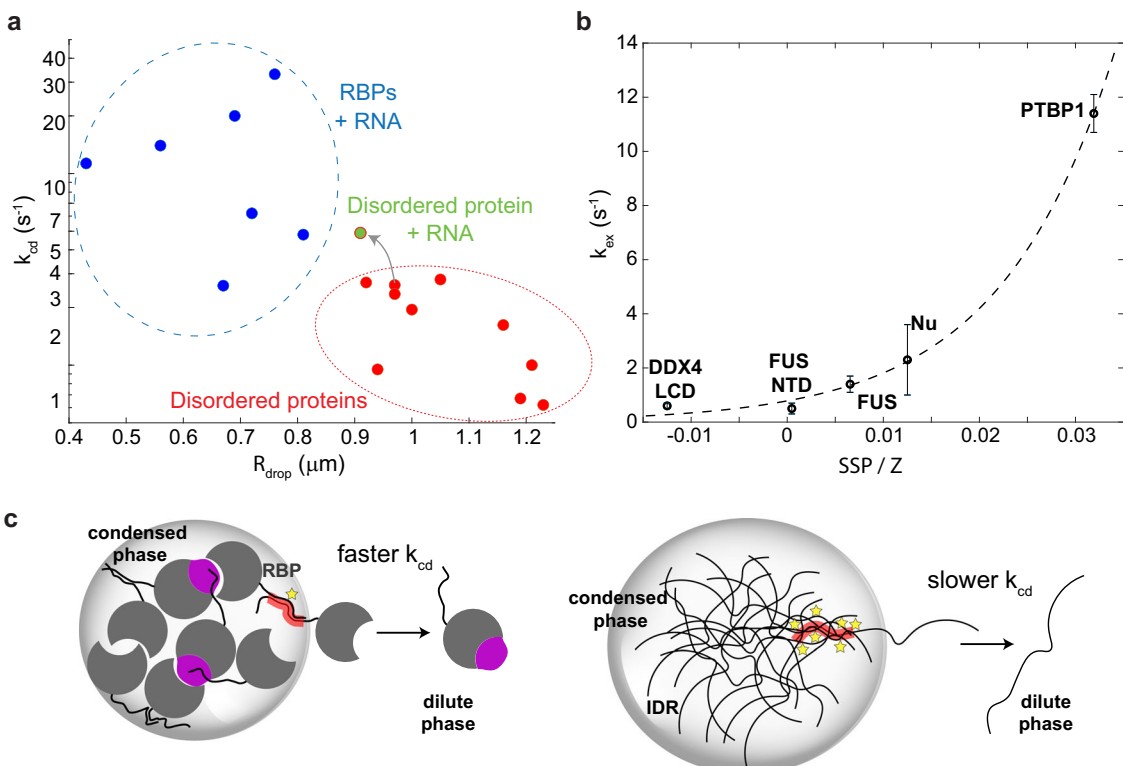

**Fig. 4 | Structured and disordered proteins exhibit distinct differences in condensate properties. a** A correlation plot of chemical exchange vs. the average radius of the droplets showing that disordered proteins make larger droplets that exchange slower to the dilute phase compared to RNA-binding structured proteins. The latter cover much broader space with much more heterogeneous interphase chemical exchange. Addition of RNA to one of the FUS samples shifted the position in the plot closer to RBPs suggesting that the RNA makes droplets more dynamic (green dot). **b** A plot of average global exchange rate determined by LLPS REDIFINE with respect to the ratio between protein secondary structure prediction factor SSP

and protein net charge Z. An obvious correlation could be deduced. Values presented for FUS NTD, FUS and Nu condensates are the averages with standard deviation of the results from multiple (n > 3) different samples containing these proteins while for DDX4 and PTBP1 are the fitting uncertainties. **c** A potential explanation for such behavior. RBDs are mostly folded proteins that protect their hydrophobic parts with their fold and can engage in far less polyvalent binding interactions compared to IDPs. Therefore, IDPs have larger affinity to be in the condensed, interaction-reach phase. Source data are provided as a Source Data file.

structured protein condensates which phase separated in the presence of RNA (Supplementary Fig. 9). The effect of protein structural properties on droplet dynamics may find its origin by the positive correlation between the global chemical exchange rate and the ratio between the protein secondary structure prediction factor (SSP) and the protein net charge Z (Fig. 4b). The fastest exchange is observed for PTBP1 that has the largest *SSP/Z* ratio while the fully unstructured FUS NTD and DDX4 LCD domains have the slowest rate and smallest *SSP/Z* ratio (Fig. 4b). A possible explanation for this relationship is proposed in Fig. 4c that highlights the smaller potential for multivalent interactions in structured RBPs compared to fully unstructured proteins to form condensates.

### Beyond LLPS: Determination of interaction affinities in soluble complexes

The concept of restricted diffusion in biomolecular condensates occurs as a result of the droplets' interfacial tension[52]. Polyvalent interactions in the condensed phase prevent molecules from diffusing to the dilute phase, manifesting as complex diffusion. Interestingly, even dispersed molecular complexes experience restricted diffusion. Being in dynamic equilibrium with the complex, constituent molecules move in solution as they are constantly being dragged by these intermolecular interactions, which makes their diffusion coefficient a function of interaction affinity and kinetics. Diffusion NMR experiments are sensitive to detect this apparent restricted diffusion only when diffusion coefficients of free molecules and complex are significantly different and in an intermediate exchange regime (10–1000 ms). Fortuitously, this is often the case with protein-RNA complexes prone to phase separation, resulting in complex, exchange-averaged diffusion detectable by the NMR experiment.

Considering this, we extended our model with a multi-pool exchanging system in a dilute phase. This is illustrated in Fig. 5a; if the diffusion coefficient of the complex is significantly different from that of the free species, then REDIFINE could, in principle, determine the binding kinetic parameters such as the off-rate $k_{off}$ and the population of the complex. This can be used to further calculate the dissociation constant $K_d$ and the on-rate $k_{on}$ ("Methods"). We applied the methodology to soluble PTBP1 complex with one equivalent of 3 x UCUCU RNA (Fig. 5b). Using this extended model, we obtained a population for the complex of $0.945 \pm 0.006$ and a $k_{off}$ of $1.41 \pm 0.02\,s^{-1}$. Based on the previous experience with this system, we presumed a 1:1 binding stoichiometry and could determine a $K_d$ of $500 \pm 100$ nM. This value is in very good agreement with the independent electrophoretic mobility shift assay (EMSA) performed for the same complex with a measured $K_d$

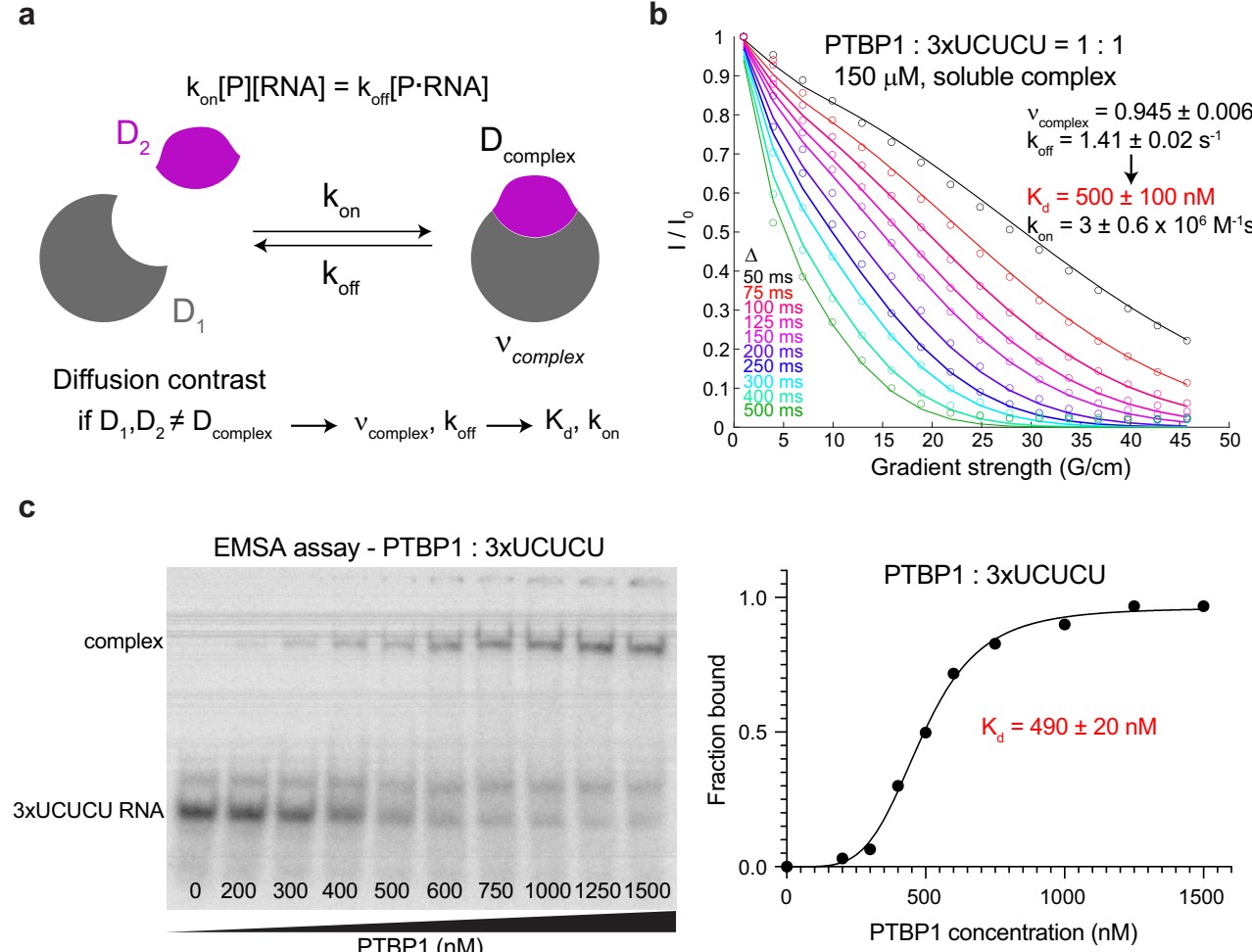

**Fig. 5 | Application of REDIFINE beyond LLPS. a** REDIFINE with dispersed complexes. If biomolecules coexist in solution in multiple exchanging states with different diffusion coefficient (protein/RNA and a complex), then PGSTE curve contains exchange averaged information of all the states present in solution. If exchange happens to be in intermediate regime, REDIFINE approach can be used to calculate $K_d$ and $k_{on}$ rate based on fitted $k_{off}$ and the complex population parameters. **b** Fitting of REDIFINE data for PTBP1 : 3 × UCUCU. $K_d$ of $500 \pm 100$ nM is determined. Values of the parameters are the average values of two independent measurements and two independent processing (different integration regions) with the standard error. **c** Native gel showing the 3 × UCUCU RNA shift upon binding to PTBP1 protein and ensuing fitting of the protein-RNA complex band formation. $K_d$ of $490 \pm 20$ nM is determined which is in agreement with the REDIFINE. The $K_d$ value is the mean of three independent EMSA experiments with standard error. Source data are provided as a Source Data file.

of 490 ± 20 nM (Fig. 5c). The additional advantage of our methodology is that it involves a single protein–RNA sample for an acquisition/processing time of only 2 h and importantly does not require titration.

## Discussion

Here we presented a methodology that can be used to study liquid–liquid phase separation and complex biophysical systems in general. REDIFINE approach allowed us to gain detailed insights into the physical properties of biphasic condensates by examining them in vitro in their biologically relevant form. By determining diffusion coefficients in dilute and condensed phases co-existing in equilibrium, partition coefficients, average droplet size, surface permeability, and chemical exchange, REDIFINE provides a fingerprint for biomolecular condensates, akin to fluorescent microscopy LLPS assays, FRAP, and microrheology combined[25], though acquired in a single NMR experiment and without requiring any specific detection labels. As NMR spectroscopy provides distinct molecular fingerprints, LLPS REDIFINE can be applied to any species present in the condensates, given that they are observable by NMR, i.e., in multicomponent condensates. Although we primarily focused on monitoring proteins (FUS, DDX4, PTBP1, and SARS-CoV-2 Nu), we could also observe the properties of RNA and water within the condensates. This multicomponent characterization, often not accessible by other techniques, can be extended to ions such as phosphate, chloride, and sodium[53] that often influence phase separation behavior, but also multiprotein and drug-protein condensates.

Applying LLPS REDIFINE on various protein condensates, we observed an unexpected trend in droplet properties. We could see an inverse correlation between the droplet size and the chemical exchange between the condensed and dilute phases for our samples. This might be explained by the fact that smaller droplets have a larger surface-to-volume ratio promoting more efficient exchange processes[54]. Moreover, RBPs containing structured domains tend to form smaller and more dynamic droplets compared to mostly unstructured proteins. We hypothesize that this is due to the much smaller valency of structured RBP homotypic interactions compared to IDPs. This is also why structured proteins need an anionic RNA to significantly increase the polyvalency of interactions in order to enable phase separation, while IDPs can engage in interactions with other protein molecules and phase-separate on their own. Once in the droplets without tertiary structure protecting their hydrophobic or other interaction-prone groups, IDPs have a higher affinity to remain in the high-valency droplet environment compared to the dilute phase. This would explain consistently slower exchange rates among IDPs.

LLPS REDIFINE requires for NMR analysis only one sample of unlabeled biomolecules stabilized in agarose hydrogel. We performed an extensive analysis to show that the loose mesh of 0.5% agarose does not affect the structure and behavior of the large biomolecules of interest (Supplementary Note 6). A minimum of 100-150 μM protein in 130 μl sample volume is needed for analysis, although even lower protein concentrations can be used since NMR experiments can be acquired with more extensive signal-averaging. As the spectral resolution is not essential, LLPS REDIFINE is in theory also applicable at low-field benchtop machines with gradient capabilities.

Currently, LLPS REDIFINE is based on a model that considers droplets of uniform size. Consequently, it provides information about the average droplet radius that best fits the experimental data. Although agarose-mimicking cytoskeleton prevents droplet fusion, in general, the droplets in the biphasic sample are somewhat heterogeneous in size. Although not encountered in this study, very heterogeneous size distributions can affect the fitting. Principally, if the distribution shape is assumed (normal or log-normal) LLPS REDIFINE could provide also a standard deviation of the distribution besides the average radius[55–58] and this is currently being investigated. It should be mentioned that very fast exchange could also constrain the

applicability of REDIFINE, which is, in principle, limited to the kinetic processes occurring on a millisecond to second time scale. Other limitations are discussed in the Supplementary Information. Furthermore, while the model currently accounts only for the surface-to-volume ratio of the droplets, it is also possible to include different liquid droplet shapes, other than spheres. This could be ellipsoids and cylinders for example[38], which could be applicable in the case of aggregated condensates and also for in cell (bacterial and eukaryotic) studies, which are currently being explored.

Biphasic condensates exhibit complex diffusion behavior that manifests as a deviation from mono-exponential trend and resembles the so-called kurtosis effect[59] utilized traditionally in Magnetic Resonance Imaging (MRI). This kurtosis effect, defined with a parameter $K$, represents a deviation from gaussian diffusion (Supplementary Note 7) and implies a physically-restricted molecular motion[36]. Diffusion kurtosis imaging of water has proven to be very sensitive to the tissue type and is conventionally used to assess structural abnormalities in the brain associated with neuropathologies. It is illustrated in Supplementary Note 7 that molecular diffusion in all biomolecular condensates experiences a kurtosis effect to a different extent. Following this link, we anticipate that REDIFINE methodology can be eventually utilized for in cell NMR microimaging[60] to spatially characterize various membraneless assemblies.

Additionally, this methodology exhibits a broad range of applications, such as for soluble biomolecular complexes, as it can provide thermodynamic and kinetic information of protein-RNA binding in the intermediate to fast exchange regime. As the diffusion coefficient of a small molecule becomes significantly slower upon binding to a large biomolecule, we anticipate that our approach can be exploited in drug screening campaigns as ligand-observed experiments.

Altogether, LLPS REDIFINE provides a label-free approach to study droplet properties in a biphasic environment for protein-only condensates but also for homogenous multicomponent condensates composed of protein and RNA. Application to multiprotein and protein-drugs condensates is underway. Considering the importance of the surface of biological condensates for droplet maturation and reaction kinetics[61–65], the exchange parameters obtained by LLPS REDIFINE will be crucial to understand the role of condensates in biology and disease.

## Methods

### PGSE NMR experiments to probe translation diffusion

Since the early development of the pulsed gradient spin echo (PGSE) technique by Stejskal and Tanner in 1965[66], diffusion measurement experiments have been extensively used in the field of NMR and MRI[67–69]. In the conventional diffusion experiment, a normalized signal of a studied analyte is mapped as a function of the gradient amplitude applied in the experiment. The application of gradients can impart a specific phase to molecules depending on their position in the NMR tube, and their translational diffusion will lead to the mixing of different phases, ultimately resulting in decay of the signal. The attenuation of the observable normalized signal depends on the experimental parameters and diffusion rates (Supplementary Note 1) and follows a simple mono-exponential decay[66] according to the formula

$$\frac{I}{I_0} = e^{-(\gamma^2 g^2 \delta^2 \cdot \Delta) \cdot D} = e^{-b \cdot D} \qquad (1)$$

where parameter $b = \gamma^2 g^2 \delta^2 \cdot \Delta$, $\gamma$ is the nuclear gyromagnetic ratio, $g$ and $\delta$ the strength and duration of the gradient coming from the NMR probe, $\Delta$ diffusion time, and finally $D$ a diffusion coefficient. It is often sensible to write this equation as

$$\frac{I}{I_0} = e^{-(\gamma^2 g^2 \delta^2 \cdot \Delta) \cdot D} = e^{-q^2 \Delta \cdot D} \qquad (2)$$

where $q = \gamma g \delta$ it excludes diffusion time and depends only on the pulse gradient parameters. The decay of the signal with respect to $\Delta$ or $\delta$ provides a quantitative measure of the displacements, and therefore, the diffusion rates of the nuclear spins along the applied gradient field can be calculated (Supplementary Note 1). Usually, $\delta$ it is kept constant, but should be adjusted according to the molecular weight of the analyte. When this dependence is visualized on a semi-logarithmic plot as

$$\ln \frac{I}{I_0} = f(b) = -bD; \quad \ln \frac{I}{I_0} = f(q^2) = -q^2 \Delta \cdot D \quad (3)$$

Then, a slope of the line simply represents the diffusion coefficient (Supplementary Note 1).

## Mathematical foundation of the LLPS REDIFINE model

Laying the foundations of LLPS REDIFINE (REstricted DIFusion of INvisible speciEs) methodology, the model considers a biphasic sample containing a relative population of protein $\nu_{cond}$ in condensed and $\nu_{dil}$ in dilute phase. For simplicity, we considered a mono-component condensate formed solely by one type of biomolecule (i.e., FUS condensates). Since the chance for hindered diffusion outside the droplets is very low due to the relative volumes of condensed and dilute phase, we consider free diffusion of biomolecules in the dilute phase described with the diffusion coefficient $D_{dil}$. Diffusion of molecules inside the condensed droplets is restricted by the interfacial tension caused by the very different interaction affinity within the two phases. Related NMR diffusion measurements were first applied in the 2000s to characterize emulsion systems based on restricted diffusion[55–58]. In biomolecular condensates, the polyvalent interactions inside the condensed phase prevent the molecules from diffusing freely to the dilute phase. This defines certain permeability[38,70] of droplet interface described with permeability factor p. Restricted diffusion will depend on the size of the droplets, and we assume perfect spherical geometry for the condensed droplets defined by the average radius $R_{drop}$. For the sake of simplicity, if we use $q = \gamma g \delta$, the restricted diffusion[36–38,70–73] can be expressed as:

$$D_{cond}^{app} I = -\frac{1}{b} \ln \left[ 2 \frac{1 - \cos(qR)}{(qR)^2} + 4(qR)^2 \sum_{n=1} e^{(-n^2 \pi^2 D_{cond} \frac{\Delta}{R^2})} \frac{1 - (-1)^n \cos(qR)}{\left((qR)^2 - (n\pi)^2\right)^2} \right] \quad (4)$$

For the fitting purposes, we propagated the summation up to the first 2000 terms.

To describe the chemical exchange between the condensed and dilute phase, we used the exchange rate $k_{cd}$ that is dependent on the droplet surface-to-volume ratio and interface permeability. As the surface-to-volume ratio for the spheres is equal $\frac{3}{R_{drop}}$, then the chemical exchange rate[38] is given by:

$$k_{cd} = \frac{3}{R_{drop}} p \quad (5)$$

We can use the McConnell differential equations to describe the chemical exchange between the two phases. If the decay of magnetization M due to the gradients of PGSE experiment can be expressed in the differential form as:

$$\frac{dM}{dt} = -q^2 D \cdot M \quad (6)$$

Then the evolution of the biphasic system in exchange can be shown as:

$$\frac{d}{dt} \begin{pmatrix} M_{cond} \\ M_{dil} \end{pmatrix} = \begin{pmatrix} -q^2 D_{cond}^{app} - k_{cd} & k_{dc} \\ k_{cd} & -q^2 D_{dil} - k_{dc} \end{pmatrix} \begin{pmatrix} M_{cond} \\ M_{dil} \end{pmatrix} \quad (7)$$

where $k_{dc} = \frac{\nu_{cond} k_{cd}}{\nu_{dil}}$ and $\nu_{cond} + \nu_{dil} = 1$ as the system is at equilibrium during rather short measurement time. We can also define a global chemical exchange as a rate more comparable with the rates obtainable by FRAP experiments.

$$k_{ex} = \nu_{cond} k_{cd} = \nu_{dil} k_{dc} \quad (8)$$

The system of differential equations in Eq. (7). can be solved analytically, and the total magnetization at given time $t = \Delta$, and gradient strength and length g and $\delta$, respectively, is given by:

$$S(q^2, \Delta) = M_{cond}(q^2, \Delta) + M_{dil}(q^2, \Delta) \quad (9)$$

Solution for $S(q^2, \Delta)$ has been used to fit the experimental data. There are 5 independent physical parameters that describe the model system: $D_{dil}$, $D_{cond}$, $\nu_{cond}$, $R_{drop}$, and average permeability p. The diffusion decay curves for various $\Delta$ have been fit globally at the same time. Fitting was performed using fminsearch, a nonlinear programming solver in MATLAB, that finds the minimum of multivariable function. The function to be minimized was defined according to the least squares regression as the Euclidian norm of the difference between the experimental data points and model predictions.

$$\chi = \sqrt{\sum_{i=1}^{j} \left| \frac{I}{I_0} - S(D_{dil}, D_{cond}, \nu_{cond}, R_{drop}, p) \right|^2} \quad (10)$$

The uncertainties of fitted values were calculated using the covariance matrix of parameters. Covariance matrix reports on the correlations between all the parameters and therefore reflects on the ambiguities about their fitted values. In other words, large uncertainty reports that the parameter is not well correlated with the other parameters and therefore cannot be determined with certainty by the multiparametric fitting procedure. For example, this is the case with diffusion coefficient $D_{cond}$ in case of low population of the condensed phase (<0.2) or fast chemical exchange (>30 s$^{-1}$).

The model to calculate binding affinity in soluble complexes considers a two-pool system with different isotropic diffusion coefficients (as in a dispersed sample) that can exchange with each other. There are 4 independent parameters that describe the model system: $D_{dil\_free}$, $D_{dil\_complex}$, $\nu_{complex}$, and $k_{off}$. Based on known initial concentrations of the reactants, the population in the bound state ($\nu_{complex}$), and a priori determined binding stoichiometry, one can calculate the equilibrium concentrations of the species in a free state. Having all the equilibrium concentrations, a dissociation constant can be determined as

$$K_d = \frac{\nu_{1,free} \nu_{2,free}}{\nu_{complex}} \quad (11)$$

Based on the $k_{off}$ and $K_d$, one can also calculate the $k_{on}$ rate of complex association:

$$k_{on} = \frac{k_{off}}{K_d} \quad (12)$$

## Protein purification

Full-length Nucleocapsid construct was cloned into a pESPRIT vector between the AatII and NotI cleavage sites with His6-tag and TEV protease cleavage sites at the N terminus (GenScript Biotech, The Netherlands). All Nucleocapsid protein constructs were expressed in *Escherichia coli* BL21 (DE3) overnight at 18 °C after induction at an optical density of 0.6 with 0.6 mM IPTG. Cells were harvested by centrifuging at 3,750 x g and resuspended in buffer containing 20 mM Tris-HCl, pH 8.0 and 1 M NaCl. The cells were lysed by cell cracking and the lysate centrifuged again at 17,000 × g at 4 °C. The supernatant was subjected to standard Ni-NTA

purification. Proteins were eluted with 20 mM Tris, pH 8, 500 mM NaCl, and 300 mM imidazole. Samples were then dialyzed against 20 mM Tris, pH 8, 300 mM NaCl, and 5 mM 2-mercaptoethanol at 4 °C overnight. Following TEV cleavage and removal of the excess N-terminal tag and TEV by Ni-NTA affinity, samples were concentrated and exchanged to 20 mM sodium phosphate, pH 6.0, 50 mM NaCl (NMR buffer). For certain protein preparations, when purity was not satisfactory, it was additionally subjected to size exclusion chromatography (SEC; Superdex 75/200) into the NMR buffer.

The coding sequence of PTBP1 was cloned in pET28a (Novagen). All plasmids were sequenced and transformed into BL21-Codon Plus (DE3)-RIL cells (Agilent Technologies) for protein expression. Protein purification of PTBP1 was performed as described previously[74]. The NMR samples of PTBP1 were prepared in 10 mM sodium phosphate buffer with 20 mM NaCl at pH 6.5.

Full-length FUS sequence was cloned into a pET24b vector by XhoI and BamHI restriction digestion and ligation that enables the translation of a TEV-cleavable GB1-His6-tagged protein. RNA-binding mutants were generated by classical site-directed mutagenesis. FUS protein constructs were expressed in *Escherichia coli* BL21(DE3) overnight at 20 °C after induction at an optical density of 0.6 with 0.1 mM isopropyl-β-d-thiogalactopyranoside (IPTG). Cells were harvested by centrifugation at 5,000 ×g and were directly resuspended in Suspension Buffer (50 mM HEPES, pH 7.5, 150 mM NaCl, protease inhibitors). Two times 12 min sonication on ice for 7 cycles with pulse-pause intervals at power level 70% was used to lyse the cells. The cell lysate was centrifuged at 15,000 ×g at 4 °C, and the resulting cell pellet was homogenized on ice (15 mL, Dounce Homogenizer) in Buffer A (8 M urea, 50 mM HEPES, pH 7.5, 500 mM NaCl). The homogenized protein solution was centrifuged again at 15,000 ×g at 4 °C for 25 min. The supernatant was collected and subjected to Ni-NTA purification. The Ni-NTA (Qiagen) column was equilibrated with Buffer A. After loading the filtered protein solution onto the column, the column was washed with 2 column volumes of Buffer A and 4 column volumes of Buffer A1 (1 M urea, 50 mM HEPES, pH 7.5, 150 mM NaCl). Protein was eluted by Elution Buffer (1 M urea, 50 mM HEPES, pH 7.5, 150 mM NaCl, 250 mM imidazole). TEV protease and 5 mM 2-mercaptoethanol were added into the eluted protein solution followed by a dialysis in Buffer B (1 M urea, 50 mM HEPES, pH 7.5, 150 mM NaCl, 250 mM imidazole, 5 mM ME) overnight, prior to a second dialysis in Buffer C (6 M urea, 50 mM HEPES, pH 7.5, 150 mM NaCl). A second Ni-NTA purification was performed by applying the protein solution on a Ni-NTA column equilibrated in Buffer C. Six column volumes of Buffer C were used for washing. The flow-through of protein solution and washes were collected together and concentrated via centrifugation using 15 mL Amicon UltraCentrifugal Filter Units (Millipore) with a molecular weight cut-off of 30 kDa. After dilution of high-concentration urea to form the droplet sample, the final buffer contained 5 mM phosphate and 5 mM HEPES at pH 7, 1 mM TCEP, 600 mM urea, and 15 mM NaCl.

N-terminus disordered domain of DDX4, DDX4(1–233), and FUS NTD low-complexity domains were expressed and purified as previously described[19]. The final buffer used in LLPS REDIFINE was 30 mM HEPES with 200 mM KCl at pH 7.5 with a residual urea concentration of 0.6 M. For reasons unrelated to the present study FUS NTD (600 μM sample in Fig. 1) was $^{15}$N/$^{13}$C labeled.

## RNA purification
S2m (5′-GGUUCACCGAGGCCACGCGGAGUACGAUCGAGUGUACAGU GAACC-3′), 3 × UCUCU (5′-GGGAGAUCUCUAAAAAUCUCUAAAAAUCU CUAAAAA-3′), and SL34 (5′-GGGAUCCGAUUUCCCCAAAUGUGGGAAA CUCGACUGCAUAAUUUGUGGUAGUGGGGGACUGCGUUCGCGCUUU CCCCUG-3′) RNAs were produced by in vitro run-off transcription with T7 RNA polymerase (purified in-house) from two complementary DNA primers containing a T7 promoter or in the case of SL34 based on previously reported plasmids[75]. Magnesium concentration was

optimized for in vitro transcription reactions with both commercially available unlabeled NTPs (Applichem). The RNAs were purified by anion exchange chromatography in denaturing conditions. The purified RNA was precipitated by butanol extraction to eliminate urea and salts. Lyophilized RNA was re-suspended in water or respective buffers. A6 RNA oligo was purchased from Dharmacon.

## Turbidity measurements
The turbidity (light scattering at 600 nm) of the samples was measured using a UV-Vis spectrophotometer (ND-1000 Spectrophotometer, NanoDrop). For Nu, a defined volume of protein and RNA stock solutions was diluted in Nucleocapsid NMR buffer (20 mM sodium phosphate, pH 6.0, 50 mM NaCl) to yield a 60 μM of protein and varying RNA concentrations in a total reaction volume of 20 μL. The RNA is added in the last step to initiate the phase separation, and the measurements are performed after 1 min of incubation in two technical replicates. In total, three biological replicates were acquired using the same procedure for Nu:s2m and two biological replicates (each with 2 technical replicates) for Nu:A6. A similar procedure was used for PTBP1 and SL34 (3 biological replicates) prepared at 100 μM protein concentration in PTBP1 buffer (10 mM sodium phosphate, pH 6.5, 20 mM NaCl) in a total reaction volume of 10 μL. For measurements of FUS, 20 μM protein samples were prepared in FUS Turbidity Buffer (5 mM potassium phosphate, pH 6.0, 100 mM KCl, 2 μM ZnCl$_2$, 1 mM TCEP) and measured in three replicates with varying concentrations of RNA.

## Condensate preparation in agarose gel
All biphasic samples examined in this study with LLPS REDIFINE method were prepared in 0.5% final mass concentration of agarose in order to stabilize liquid droplets as previously described[19]. For structured protein, low-melting agarose is used while for unstructured proteins we used a regular agarose. Corresponding protein buffers were first boiled with agarose powder to solubilize it, and then kept warm to remain liquid. Agarose buffers used to prepare FUS and DDX4 samples were still liquid at 55 °C, while the low-melting agarose buffer is in liquid state even at 36 °C. It is crucial to use low-melting agarose with folded proteins in order not to denature them by the addition of very hot agarose stock buffer. To form a biphasic sample stabilized by agarose hydrogel, protein (and RNA) stocks were mixed with respective warm agarose buffer in a 1.5-ml Eppendorf and quickly transferred to the NMR tube. Agarose gelation occurs shortly after transferring to the NMR tube, and the sample is ready for analysis. Note that the final samples of FUS and DDX4 also contain residual urea[19].

## Microscopy
FUS NTD (100 μM), full-length FUS (150 μM), Nucleocapsid:A6 (110:660 μM), and PTBP1:3 × UCUCU (150:15 μM) samples were prepared in 0.5% agarose hydrogel to ensure the same conditions as in REDIFINE experiments and imaged using bright-field microscopy. FUS NTD, FL FUS and Nucleocapsid protein were labeled post-translationally with Atto488 dye (Jena Bioscience) on primary amino groups as per the manufacturer's protocol. Unbound dye was removed from the labeled protein by overnight dialysis against fresh protein buffer. GFP- and mCherry-labeled FL FUS were expressed as fusion proteins and expressed as described above for FUS FL. S2m RNA was 3′end labeled with pCp-Cy5 (Jena Bioscience) using T4 RNA ligase (NEB). Enzymes and unbound dye were removed by phenol-chloroform extraction and ethanol precipitation. Unlabeled FUS NTD and FUS FL were spiked 100:1 with Atto488 and GFP/mCherry-labeled protein. Unlabeled Nucleocapsid and s2m RNA were spiked 20:1 with Atto488 Nucleocapsid or Cy5-s2m, respectively. All samples except for Nucleocapsid:s2m were prepared in corresponding buffers containing 0.5% agarose. FUS NTD, FUS FL, Nucleocapsid:s2m and PTBP1:3xU-CUCU samples were imaged on a Nikon Spinning Disk SoRa microscope using a 100 × 1.45 CFI Plan Apo Oil objective, while

Nucleocapsid:A6 image was acquired using a Sony α6400 APS-C DSLM camera body mounted to an Olympus CKX41 inverted microscope with a 40× objective. FRAP experiments were performed using a standard procedure[76]. Ensuing recovery rates are reported as a multiple droplets average (>4) with standard deviation.

## EMSA assay

Purified RNA 3 × UCUCU (10 pmol) was first 5′-end dephosphorylated by 5 units of Antarctic phosphatase (NEB, 5 U/μL) in Antarctic phosphatase buffer (NEB, 50 mM Bis-Tris-Propane-HCl, 1 mM MgCl$_2$, 0.1 mM ZnCl$_2$) in a total reaction volume of 10 μL. The reaction was incubated at 37 °C for 30 min, and the enzyme was heat-inactivated at 80 °C for 2 min Dephosphorylated RNA (1.8 pmol) was then 5′-end labeled with 2 pmol of ATP [γ$^{32}$P] (10 uCi/uL, 5000 Ci/mmol, Hartmann Analytic GmbH) using 10 units of T4 polynucleotide kinase (NEB, 10 U/μL) in T4 polynucleotide kinase buffer (NEB, 70 mM Tris-HCl, 10 mM MgCl$_2$, 5 mM DTT) in a total reaction volume of 10 μL. The reaction was incubated at 37 °C for 1 h, and then the enzyme was heat-inactivated at 95 °C for 5 min. The labeled RNA was purified using an Illustra MicroSpin G-25 column (GE Healthcare) and diluted to 7.2 fmol/μL. Different protein-RNA samples containing 7.2 fmol 3 × UCUCU and an increasing amount of PTBP1 full-length (final concentration from 0 to 1.5 μM) were prepared in D buffer (20 mM HEPES, pH 8.0, 100 mM KCl, 1.5 mM MgCl$_2$, 10% glycerol, 0.5 mM DTT). The RNA-protein complexes were run on a native 10% TBE gel in TBE running buffer (40 mM Tris-HCl, pH 8.3, 45 mM boric acid, 1 mM EDTA) at 100 V for 1 h. The gel was soaked in fixing solution 1 (10% acetic acid, 20% methanol, 5% glycerol) for 15 min and in fixing solution 2 (20% methanol, 5% glycerol) for another 15 min. The gel was subsequently dried using a drying device (Hoefer Slab Gel Dryer GD2000) at 80 °C under vacuum for 1 h. The gel was screened overnight in a phosphor cassette and imaged using Typhoon FLA 9000 (GE Healthcare).

## NMR spectroscopy

All NMR experiments were recorded with a 3-mm NMR sample tube at 298 K (unless otherwise specified) on 600 MHz and 700 MHz Avance NEO spectrometers equipped with TCI cryo-probes and on a 500 MHz Avance NEO with QCI cryo-probe. In total 700 MHz cryoprobe was equipped with xyz-gradient system, while the 500 and 600 MHz probes had only z-axis gradients. All experiments were run and processed using TopSpin 4 software (Bruker).

LLPS REDIFINE data sets were measured using a conventional 2D Stimulated-Echo experiment using bipolar gradients (stebpgp1s19) to which a pseudo 3$^{rd}$ dimension was added to vary the diffusion encoding time (Supplementary Note 3). Watergate3919 was used as a preferred water suppression scheme. Alternatively, we also tested the LED experiment using bipolar gradients (ledbpgppr2s) and water suppression using presaturation, and, besides lower signal-to-noise (SNR), the same LLPS REDIFINE results were obtained. LLPS REDIFINE of the water molecule in the FUS condensate (Fig. 2) was acquired without water presaturation using ledbpgp2s pulse sequence.

All diffusion experiments were run in 16 increments of gradient strength (g), ranging from 2 to 95%. On Bruker spectrometers, this is defined in percentage of the maximum gradient amplitude of the system, and it is specific for each NMR probe. Typically, most probes can deliver a maximum output of around 53.5 G/cm at 100% gradient amplitude. LLPS REDIFINE needs to contain at least 16 gradient amplitude points defining the diffusion decay for a good fitting. 16−32 scans of signal averaging to acquire sufficient SNR per every gradient strength were required on our NMR systems, and condensate samples were prepared at 100−150 μM protein concentration (except FUS NTD at 600 μM). Depending on the protein/RNA size, diffusion gradient length (δ) can range from 2 to 10 ms. In this study, we used δ of 9 or 10 ms for the protein detection, 3.5 ms for the A6 RNA, and 2 ms for water, and always smoothed rectangular-shaped gradients SMSQ10.100. All experiments were run using an AU program "dosy"

available in the TopSpin that also calculates the gradient ramp and feeds it to the spectrometer.

It is described that the LLPS REDIFINE data set contains multiple diffusion curves acquired for different diffusion encoding times. To provide the most accurate multiparametric fitting, a minimum of six diffusion curves have to be acquired spanning as wide as possible diffusion encoding time. The maximum diffusion time that can be applied is constrained by the longitudinal T$_1$ relaxation of the target molecule, leading to the signal decay, and for RBDs it is around 700 ms, while for unstructured proteins it can be up to 1000 ms. In case of slow chemical exchange, long diffusion times are absolutely crucial to resolve the fitting parameters. As chemical exchange is not a priori known, acquisition of sufficient diffusion curves spanning the region between the short and the maximum diffusion time is acquired for precise parameter determination by LLPS REDIFINE. In this work, we acquired from 6 to 11 curves in total that were used for fitting. The reason for acquiring fewer diffusion decay curves can be the need to acquire the data faster due to the instability of the sample and/or limited experimental time. Also, samples containing similar fractions of protein in the condensed phase and dilute phase can be characterized well with only 6 curves, while more curves are required otherwise. Further aspects of the REDIFINE methodology, including prospects and limitations, are discussed in Note 8 of the Supplementary Information.

Another parameter worth mentioning is the d$_1$ recovery delay that we used between 1.5 and 5 s. To acquire the full LLPS REDIFINE data set, the course depends on all aforementioned parameters, but in this study, it usually took 1−2 h.

## LLPS REDIFINE processing and fitting

The presented differential equations defining the model behind LLPS REDIFINE are incorporated within a home-written MATLAB code. Differential equations are first solved and their analytical solution is used for all the simulations (Supplementary Note 2) and all the fitting. The experimental data were first processed in TopSpin (Fourier transformation, apodization, baseline correction) and then fed to the MATLAB code, which used the least squares regression to minimize the difference between the experimental data and the analytical model function. Five independent parameters were fit: $D_{dil}$, $D_{cond}$, $v_{cond}$, average $R_{drop}$, and $p$, while $k_{cd}$ is derived as explained in Eq. (5). Because of using analytical solutions for the differential equations presented in Eq. (7), the multiparametric fitting procedure was very fast and lasted only a couple of seconds on a regular laptop. The uncertainties of fitted values were calculated using the covariance matrix of parameters. The code containing the model functions and fitting procedures is provided as Source data.

## Reporting summary

Further information on research design is available in the Nature Portfolio Reporting Summary linked to this article.

## Data availability

Unless otherwise stated, all data supporting the results of this study can be found in the article, supplementary, and source data files. All raw NMR data presented in this work have been deposited to Zenodo repository https://doi.org/10.5281/zenodo.15228787. Source data are provided with this paper.

## Code availability

The Matlab codes used for simulations and fitting have been deposited in the same Zenodo repository https://doi.org/10.5281/zenodo.15228787.

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

## Acknowledgments

The authors acknowledge Prof. Lucio Frydman, Prof. Thomas Michaels and Dr. Fred F. Damberger for useful discussions. The authors also thank Drs. Uwe Schmitt, Tarun Chadha, and Agnieszka Ilnicka from ETH IT Services for constructive advice that helped speed up the fitting procedure in Matlab. This work was supported by Swiss National Science Foundation (SNSF) grants 310030-215555 (F.H.-T.A.), 4078P0_198253 (F.H.-T.A.), CRSII5_205922 (F.H.-T.A.), 205321_204920 (F.H.-T.A.), and NCCR RNA & Disease grant 51NF40-182880 (F.H.-T.A.).

## Author contributions

M.N. and L.E. devised the project; M.N. designed the theoretical model and the experimental procedure; M.N. performed the simulations and experiments, as well as process the data; M.N., L.E. and F.H.-T.A. analyzed the data; M.N., Y.H., Y.N., N.K., and L.E. prepared protein and RNA samples. Y.N. performed EMSA assays. M.N., N.K., and L.E. performed microscopy imaging. M.N. wrote the first draft of the manuscript. M.N., L.E. and F.H.-T.A. reviewed and edited the manuscript. F.H.-T.A. provided supervision and acquired funding. All authors read and approved the final manuscript.

## Competing interests

The authors declare no competing interests.
