## [Transparent Peer Review file · Nature Communications]

LLPS REDIFINE allows the biophysical characterization of multicomponent condensates without tags or labels

Corresponding Author: Professor Frédéric Allain

Version 0:

Reviewer comments:

Reviewer #1

(Remarks to the Author)

Liquid-liquid phase separation (LLPS) is an important phenomena that plays a pivotal role in many cell biology processes so studying bimolecular condensates formed in such process is important. In the present manuscript, the authors provide a diffusion NMR-based method, termed by the authors as REDEFINE, for studying protein, RNA and water in bimolecular condensates. The REDEFINE method allows to characterize, non-invasively, biphasic systems providing, simultaneously, the diffusion coefficients in both phases, the fractions of the species, the average radius of the droplets their permeability from which the in-out exchange rate was estimated allowing the estimation of the global exchange rate. Importantly, all this information was obtained on NMR spectrometers equipped with conventional gradient systems. In fact, REDIFINE appears to use the biphasic model of Price from 1998 where the non-Gaussian (restricted) diffusion within the droplet is treated differently and is described by a kurtosis analysis as advanced by Helpert & Jensen about 20 years ago when studying structural changes in neuronal tissues. So, one may argue if the novelty imbedded in the so-called REDEFINE method is enough to surpass the novelty standard of Nature Communications. I am in the opinion that it still does.

The presented method is of high significance to the field and allows to characterize, non-invasively, multicomponent condensates which are characteristics of LLPs, which in turn, are important phenomena in cell biology. This can be performed totally non-invasively and on conventional NMR spectrometers equipped with conventional gradient systems (producing gradient pulse of ~50 Gauss/cm). In addition, the described technique may, in fact, be applied to other fields. Note however that the REDEFINE method appears to work on relatively simples systems. With the addition of more components, compartments and significant size distribution, the REDEFINE method would probably fail as mentioned in Section 5 of the Supporting Information. However, its importance arise, inter alia, from the fact the REDEFINE use no tags or labels and is completely non-invasive. This is important since the paper clearly demonstrates, as was already reported, that even a small amount of fluorescence tags might have dramatic effects on the condensates' characteristics. The flexibility of the REDEFINE method enabled the authors to study a relatively large number of different systems using the same methodology. This enabled them to identify that disordered proteins form, in general, larger condensates characterized by slower interface exchange as shown in Figure 4a. A tentative explanation is given in Figure 4.

The results of the different experiments performed support the conclusions and claims of the manuscript. The work was executed with high standards and is very well presented. The authors has also performed control experiments and did the requested verification. For example, the extracted REDEFINE droplet size was compared with the results obtained from microscopy, while the exchange rate extracted from the method was compared to the results of FEXSY experiments performed on the same samples (Extended Data Figs. 5 and 7). There are no real flaws in the data analysis, the interpretations and conclusions of the methodology presented are sound and very well described and after the deposition of the REDEFINE MATLAB code, I believe that different researchers will be able to implement the method.

So overall, the paper describes an important technique that may have large implications in LLPS and bimolecular condensates research and in cell biology research in general. I am therefore in the opinion that the manuscript can be accepted for publication in Nature Communications; however, there are few, mostly minor issues that need to be addressed before publication is granted:

1) It will be nice to repeat the measurement (say three times) on the same system and repeat the analysis to demonstrate the in sample REDEFINE reproducibility. In addition, there is a need for repeated preparation (3 or more) of one or two

representative samples and subsequent analysis to assess the sample variability, the uncertainty and variability of the different parameters extracted by the REDEFINE method.

- 2) In the diffusion signal decay graphs in the manuscript, extended data and SI, please change I/I0 to I/I0.
- 3) The authors should consider moving Figs. 5 and 7 from the extended data into the paper itself.
- 4) In fact, many references are missing. For example, in the first paragraph of the methods section a statement like “have been extensively used in the field on NMR and MRI” with no references appears odd. In recent years many models that deal with diffusion in multi-compartmental systems incorporating non-Gaussian (restricted) diffusion and exchange were developed. None of those are mentioned in the paper.
- 5) Refs. 30, 31, 41, 56, 63 and 70 should be completed, abbreviated, or corrected.
- 6) In Table 1, I think the authors should use only physically meaningful digits. For example a Radius of 1.006 micron means that the diffusion measure 6 nanometer. Do the authors think that this is indeed the case? What is the translation during the pulse gradient of 2 to 10 msec?
- 7) Caption of Fig.2 in the SI, please add the units in the last line.
- 8) In Fig. 3a in the SI, please add the TE/2 to the pulse sequence. It is important to state what were the TE/2's in the case of pulse gradient duration of 2 and 10 msec. The authors may want to comment on the effect of T2 that is active during the TE period.

Reviewer #2

(Remarks to the Author)

The study by Novakovic et al introduces an NMR-based technique, REDIFINE, for characterizing biomolecular condensates in vitro. REDIFINE allows label-free measurements of diffusion coefficients, condensate size and fraction, and molecular exchange between condensate and the dilute phases for several types of condensates formed by RNA binding proteins. In one case, protein concentrations in the condensate and the dilute phase were also reported. The authors found that condensates formed by folded proteins are smaller and more dynamic compared to condensates formed by disordered proteins.

Overall, we find the paper well-written and has a clear structure. However, the discussions on diffusion are quite confusing and several sections are overly technical for a broad readership. A label-free technique for measurements of protein concentration and exchange dynamics is valuable to the condensate field. However, the novelty of this manuscript compared to the authors' 2021 study as well as several other label-free techniques is unclear. Importantly, the stringent sample requirements in REDIFINE could significantly limit the applicability of this technique and biological relevance of the findings. Our specific comments are listed below.

1. Several label-free techniques have been developed to probe condensate properties (including condensate rheology, thus molecular diffusion), see for example a recent review: Ibrahim, Khalid A., et al. "Label-Free Techniques for Probing Biomolecular Condensates." ACS nano. Therefore, the novelty of the current study appears overstated.
2. REDIFINE requires the sample to be prepared in agarose gel, making it hard to interpret the biological relevance of the condensate size (and potentially also the exchange dynamics) reported in this manuscript. The argument on the similarity between agarose gel and cellular cytoskeleton is quite weak. Meanwhile, there is a large body of literature on the regulation of condensate size in a free aqueous environment. Additionally, the technique requires a relatively large amount of purified protein sample and is challenging to apply to in vivo studies, further limiting the applicability and biological relevance of the current manuscript.
3. Please be more specific about modifications needed for a conventional NMR machine to achieve REDIFINE measurements, especially to achieve accurate control over “gradient strength” and “diffusion time”.
4. The discussion on “diffusion time” is very confusing. The time of a diffusion process depends on the studied length scale. Therefore, comparisons between “diffusion time” in REDIFINE and FRAP are misleading.
5. The “exchange rate” between condensate and dilute phase could be a unique property provided by REDIFINE. However, it's unclear if the “exchange rate” from the current study is an independent property from the diffusion coefficients. The FRAP recovery time after bleaching a whole condensate is often used to infer diffusion coefficient within the condensate rather than the exchange rate between phases.
6. The ability to measure protein concentrations (and partitioning coefficient) in a label-free manner could be quite valuable. It would be helpful to extend this measurement beyond FL FUS condensates.
7. The results in Fig 4 are informative, how do the quantified biophysical properties relate to factors such as molecule size, charge, or other characteristics?
8. The last section on soluble complexes raises the concern about how to distinguish whether the sample being measured contain condensates or complexes/oligomers in a REDIFINE experiment.

Minor comments:

1. Please define NTD, FEXSY.
2. Please be more specific on the number of fitting parameters, fitting error, and effect of heterogeneity in condensate size on the fitting results.
3. Please provide more information on factors that affect the Kurtosis effect and their significance in the context of the current findings.

Reviewer #3

(Remarks to the Author)

Version 1:

Reviewer comments:

Reviewer #1

(Remarks to the Author)

As stated in my previous evaluation of this manuscript the presented work is interesting, important and very nicely executed and presented.

After seeing that the authors have addressed all major issues raised by me and the other reviewer I am in the opinion that the paper can be accepted for publication in Nature Communications.

Reviewer #2

(Remarks to the Author)

Reviewer #3

(Remarks to the Author)

Version 2:

Reviewer comments:

Reviewer #4

(Remarks to the Author)

I have carefully reviewed the authors' responses to Reviewer 2's comments and the corresponding revisions made to the manuscript. Overall, I find that the authors have provided a thorough and well-justified response to the key concerns raised.

- The authors have expanded their discussion on the biological relevance of LLPS REDEFINE. Their responses clearly articulate the advantages of this approach for studying multicomponent condensates in a physiologically meaningful context.
- The authors have convincingly demonstrated how their method offers a distinct advantage over existing techniques, particularly in its ability to characterize condensates non-invasively without the need for labels or tags.
- The revised manuscript includes several examples that illustrate the applicability of the method to different protein systems. This substantiates the claim that LLPS REDEFINE can provide new insights into condensate biology.
- The authors have appropriately acknowledged the potential limitations of their approach, providing a balanced and realistic assessment of its scope and constraints.

In light of these improvements, I find the manuscript to be suitably revised and well-supported by the additional data and clarifications. I do not have any further critical concerns.

17/12/2024

Institute of Biochemistry
Prof. Frédéric Allain

E-mail allain@bc.biol.ethz.ch
Address Hönggerbergring 64
8093 Zürich
Switzerland

Manuscript NCOMMS-24-59425-T

LLPS REDIFINE allows the biophysical characterization of multicomponent condensates without tags or labels

Mihajlo Novakovic^{1,*}, Nina Han^{1,2}, Nina C. Kathe^{1,2}, Yinan Ni^{1,2}, Leonidas Emmanouilidis^{1,*}, Frédéric H.-T. Allain^{1,*}

¹Department of Biology, Institute of Biochemistry, ETH Zurich, Zurich, Switzerland

²These authors contributed equally

*Corresponding authors. Email: mihajlo.novakovic@bc.biol.ethz.ch,
leonidas@bc.biol.ethz.ch, allain@bc.biol.ethz.ch

Response letter

Dear Editor,

Thank you for your email of November 22, 2024 containing the constructive comments of the referees on our manuscript “**LLPS REDIFINE allows the biophysical characterization of multicomponent condensates without tags or labels**”. As can be appreciated from our responses below, we have now prepared a revision of our manuscript addressing all the critiques and suggestions raised by the reviewers. We provided our responses **in blue font**. For completion we are also submitting a “highlighted” version of the manuscript marking all the changes we made to the text, together with the “clean” version of the paper’s main text and the SI.

Reviewer 1

Liquid-liquid phase separation (LLPS) is an important phenomena that plays a pivotal role in many cell biology processes so studying bimolecular condensates formed in such process is important. In the present manuscript, the authors provide a diffusion NMR-based method, termed by the authors as REDEFINE, for studying protein, RNA and water in bimolecular condensates. The REDEFINE method allows to characterize, non-invasively, biphasic systems providing, simultaneously, the diffusion coefficients in both phases, the fractions of the species, the average radius of the droplets their permeability from which the in-out exchange rate was estimated allowing the estimation of the global exchange rate. Importantly, all this

information was obtained on NMR spectrometers equipped with conventional gradient systems. In fact, REDEFINE appears to use the biphasic model of Price from 1998 where the non-Gaussian (restricted) diffusion within the droplet is treated differently and is described by a kurtosis analysis as advanced by Helpert & Jensen about 20 years ago when studying structural changes in neuronal tissues. So, one may argue if the novelty imbedded in the so-called REDEFINE method is enough to surpass the novelty standard of Nature Communications. I am in the opinion that it still does.

The presented method is of high significance to the field and allows to characterize, non-invasively, multicomponent condensates which are characteristics of LLPs, which in turn, are important phenomena in cell biology. This can be performed totally non-invasively and on conventional NMR spectrometers equipped with conventional gradient systems (producing gradient pulse of ~ 50 Gauss/cm). In addition, the described technique may, in fact, be applied to other fields. Note however that the REDEFINE method appears to work on relatively simple systems. With the addition of more components, compartments and significant size distribution, the REDEFINE method would probably fail as mentioned in Section 5 of the Supporting Information. However, its importance arise, inter alia, from the fact the REDEFINE use no tags or labels and is completely non-invasive. This is important since the paper clearly demonstrates, as was already reported, that even a small amount of fluorescence tags might have dramatic effects on the condensates' characteristics.

The flexibility of the REDEFINE method enabled the authors to study a relatively large number of different systems using the same methodology. This enabled them to identify that disordered proteins form, in general, larger condensates characterized by slower interface exchange as shown in Figure 4a. A tentative explanation is given in Figure 4.

The results of the different experiments performed support the conclusions and claims of the manuscript. The work was executed with high standards and is very well presented. The authors has also performed control experiments and did the requested verification. For example, the extracted REDEFINE droplet size was compared with the results obtained from microscopy, while the exchange rate extracted from the method was compared to the results of FEXSY experiments performed on the same samples (Extended Data Figs. 5 and 7). There are no real flaws in the data analysis, the interpretations and conclusions of the methodology presented are sound and very well described and after the deposition of the REDEFINE MATLAB code, I believe that different researchers will be able to implement the method.

We are grateful for the positive comments and we try to address all the points below.

So overall, the paper describes an important technique that may have large implications in LLPS and bimolecular condensates research and in cell biology research in general. I am therefore in the opinion that the manuscript can be accepted for publication in Nature Communications; however, there are few, mostly minor issues that need to be addressed before publication is granted:

- 1) It will be nice to repeat the measurement (say three times) on the same system and repeat the analysis to demonstrate the in sample REDEFINE reproducibility. In addition, there is a need for repeated preparation (3 or more) of one or two representative samples and subsequent analysis to assess the sample variability, the uncertainty and variability of the different parameters extracted by the REDEFINE method.

This is a very good point. As 1D NMR measurements are usually very reproducible and have negligible deviation we didn't include this in our original manuscript originally. We have now addressed this point and made two different FUS NTD samples (biological replicates) and repeated the REDEFINE acquisition 3 times for each sample (technical replicates). As it

can be seen in Fig. R1 of the response letter, we have summarized the fitting results. Fig. R1a illustrates that both samples have almost indiscernible 1D spectra and that they don't change during the time course of our measurements. In the same panel, one can find the average values with standard deviation for three replicates for both samples. One can see that the REDIFINE parameters are very reproducible and there are also no dramatic inter-sample variabilities. Fig. R1b,c shows individual fitting results among samples and technical replicates. This has now been reported in the revised version of Supplementary Information.

Fig. R1: REDIFINE reproducibility. a 1D spectra of two independent FUS NTD samples at different time points during measurement. There is no significant change through time as well as variability between the two samples. Inset reports the average fitting values with standard deviation for two samples, based on three technical replicates. b,c Individual fitting for 3 experimental data sets for the samples 1 and 2, reporting fitting values and the errors reporting on the correlation between parameters based on covariance matrix.

2) In the diffusion signal decay graphs in the manuscript, extended data and SI, please change I/I_0 to I/I_0 .

If we understand correctly, the suggestion is to put 0 in the subscript. This is now done in all figures.

3) The authors should consider moving Figs. 5 and 7 from the extended data into the paper itself.

As Extended Data Figures are also an essential part of the manuscript, we prefer keeping these figures as extended data to avoid cluttering the manuscript. These data are also not crucial for the general readership.

4) In fact, many references are missing. For example, in the first paragraph of the methods section a statement like “have been extensively used in the field on NMR and MRI” with no references appears odd. In recent years many models that deal with diffusion in multi-compartmental systems incorporating non-Gaussian (restricted) diffusion and exchange were developed. None of those are mentioned in the paper.

We have added additional references in the first paragraph of the Methods section (Ref 68-70). Regarding restricted diffusion, we had in the initial manuscript a total of 13 references (Initial theory, tissue studies, colloidal systems, FEXSY experiment), with several being recent (2017-2020). We now added an additional reference (VERDICT MRI – Ref. 74) that we could find in the literature on this topic.

5) Refs. 30, 31, 41, 56, 63 and 70 should be completed, abbreviated, or corrected.

The references that appeared wrongly formatted are now corrected in the revised manuscript.

6) In Table 1, I think the authors should use only physically meaningful digits. For example a Radius of 1.006 micron means that the diffusion measure 6 nanometer. Do the authors think that this is indeed the case? What is the translation during the pulse gradient of 2 to 10 msec?

This is a very good point. We reported the values obtained from the fitting procedure with uncertainties based on the covariance matrix. The very small uncertainty in the radius means that in this data set obtaining radius from the fit was not correlated at all with the other fitting parameters. However, we agree that this value is not physically meaningful. This is now corrected in the revised table.

7) Caption of Fig.2 in the SI, please add the units in the last line.

We are grateful to the reviewer for noticing this. We have added the units.

8) In Fig. 3a in the SI, please add the TE/2 to the pulse sequence. It is important to state what were the TE/2's in the case of pulse gradient duration of 2 and 10 msec. The authors may want to comment on the effect of T2 that is active during the TE period.

This is again a very good comment. Although stimulated echo provides mostly T₁-weighting, during spin-echo the magnetization is also held transverse during the period of gradient application, and the final signal will also be affected by the transverse relaxation rate. However, in all experiments, this duration is held constant throughout a dataset so its effect on the signal attenuation is uniform and can be excluded. Almost all previous studies ignored T₂-relaxation effects. Nevertheless, if T₂ relaxation of a molecule inside the condensed phase is significantly different than in the dilute phase, this can lead to a small underestimation of the protein content in condensed phase. This is actually what limits application of REDIFINE in very structured protein and RNA condensates. However, as illustrated in Figures 1,2 and S5, the linewidths of the condensed protein ¹H signals stemming from the intrinsically disordered regions are not significantly different compared to dilute protein.

This is now mentioned in the revised manuscript.

Reviewer 2

The study by Novakovic et al introduces an NMR-based technique, REDIFINE, for characterizing biomolecular condensates in vitro. REDIFINE allows label-free measurements of diffusion coefficients, condensate size and fraction, and molecular exchange between condensate and the dilute phases for several types of condensates formed by RNA binding proteins. In one case, protein concentrations in the condensate and the dilute phase were also reported. The authors found that condensates formed by folded proteins are smaller and more dynamic compared to condensates formed by disordered proteins.

Overall, we find the paper well-written and has a clear structure. However, the discussions on diffusion are quite confusing and several sections are overly technical for a broad readership. A label-free technique for measurements of protein concentration and exchange dynamics is valuable to the condensate field. However, the novelty of this manuscript compared to the authors' 2021 study as well as several other label-free techniques is unclear. Importantly, the stringent sample requirements in REDIFINE could significantly limit the applicability of this technique and biological relevance of the findings. Our specific comments are listed below.

We thank the reviewers for their opinion about our manuscript.

Regarding the differences between this study and the study from 2021, we explain in the first section of the results how far beyond we go. While this initial study could utilize NMR diffusion measurement to estimate the population of FUS protein in the condensed phase, REDIFINE provides in addition the exact diffusion coefficients, permeability and the exchange rate as well as the size of the droplets in many different systems. In addition, water and RNA in addition to protein can now be characterized.

1. Several label-free techniques have been developed to probe condensate properties (including condensate rheology, thus molecular diffusion), see for example a recent review: Ibrahim, Khalid A., et al. "Label-Free Techniques for Probing Biomolecular Condensates." ACS nano. Therefore, the novelty of the current study appears overstated.

Thank you for pointing out this paper. Although there are some label-free methods already available as summarized in this review that is now cited, this fast-evolving field requires even more diverse methodology to study the LLPS phenomenon. We want to point out again the advantages of REDIFINE: it is completely non-destructive, requires simple sample preparation, and can be run on basically any NMR spectrometer (also cheaper low-field spectrometers). Using only one sample and in a single experiment that takes only 2 hours, REDIFINE provides the information that is currently obtained by multiple independent experiments: fluorescent LLPS assays, FRAP, and microrheology that combined require multiple sample preparations and much longer time to complete. What is more, in addition to analyzing biomolecules, REDIFINE approach can probe other condensate constituents such as water, buffer, and metabolites, all of which are not or hardly accessible to other techniques.

2. REDIFINE requires the sample to be prepared in agarose gel, making it hard to interpret the biological relevance of the condensate size (and potentially also the exchange dynamics) reported in this manuscript. The argument on the similarity between agarose gel and cellular cytoskeleton is quite weak. Meanwhile, there is a large body of literature on the regulation of condensate size in a free aqueous environment. Additionally, the technique requires a relatively large amount of purified protein sample and is challenging to apply to in vivo studies, further limiting the applicability and biological relevance of the current manuscript.

The use of agarose as cytoskeleton mimic is discussed extensively in <https://doi.org/10.1038/s41589-021-00752-3>. Inherently, most biophysical techniques have

been developed to study homogenous samples. But in the field of LLPS the samples are not homogenous as they are composed of at least two macroscopic liquid phases. As long as the two phases differ in density, which is in most cases, by definition condensates cannot exist in free aqueous environment long enough. Instead, they wet the bottom of the sample. In fact, most fluorescence microscopy and FRAP imaging done on such systems report droplets sitting at the bottom of the samples. This eventually leads to not only physical interaction with the glass altering the dynamics but also significantly larger condensates. Specifically, in the case of FUS, due to the hydrophobic nature FUS droplets eventually disappear within minutes forming a thin layer of continuous condensed phase on the glass. Furthermore, as the exchange rate happening on the surface with the environment is significant in disease-related liquid-solid transition, it is important to study condensates with similar size compared to cellular counterparts.

Low signal-to-noise and the requirement of having sufficient protein in order to get observable signal is an inherent limitation of NMR, however it has not prevented in-cell NMR from developing. In our methodology we could go down to $\sim 60 \mu\text{M}$ ($130 \mu\text{L}$) of FUS and still obtain the unambiguous condensate characterization. Furthermore, as concentration in the condensed phase is typically at the order of millimoles, we don't think that the samples that we used in this study have no biological relevance. We have also performed some preliminary measurements on FUS NTD in E.coli cells, which showed that REDIFINE has a high potential to be translated to in-cell study. These aspects are currently underway.

3. Please be more specific about modifications needed for a conventional NMR machine to achieve REDIFINE measurements, especially to achieve accurate control over “gradient strength” and “diffusion time”.

No modifications are needed. Our experiments can be performed on any conventional spectrometer on a regular probe that has gradients. Nowadays, the vast majority of probes satisfy this condition. Regular calibration of gradient strengths, usually performed when installing the NMR probe is sufficient and no additional special calibration is necessarily needed. Diffusion times are easily controlled in experiments and they are executed with utmost precision. This is discussed in the Methods section.

4. The discussion on “diffusion time” is very confusing. The time of a diffusion process depends on the studied length scale. Therefore, comparisons between “diffusion time” in REDIFINE and FRAP are misleading.

We are very sorry that our discussion about diffusion time was confusing, however, we are not aware that we compared diffusion time in REDIFINE and FRAP in the manuscript. To recapitulate, when we have a complex system consisting of several exchanging species, the choice of diffusion time will influence the outcome of an NMR diffusion experiment as we always observe an effective diffusion in the sample in a single experiment. The reason behind this is the following – the dynamics of different species mix in presence of exchange which leads to a different outcome depending on the total diffusion time. Each diffusion curve that we acquire carries unique information about the system, in other words, it reports on the diffusion coefficients, exchange, populations and droplet size up to an extent that could be encoded in a given time. The diffusion time we use is inversely proportional to the exchange it can probe, so the slow exchange usually requires longer diffusion times while the opposite is true for the faster exchange. As we don't know a priori the chemical exchange. REDIFINE dataset contains curves acquired using different diffusion times to cover the broad range of exchanges.

We comment on FRAP experiments in the next point.

5. The “exchange rate” between condensate and dilute phase could be a unique property provided by REDIFINE. However, it’s unclear if the “exchange rate” from the current study is an independent property from the diffusion coefficients. The FRAP recovery time after bleaching a whole condensate is often used to infer diffusion coefficient within the condensate rather than the exchange rate between phases.

The REDIFINE probes so-called permeability p of the droplet interface that represents effectively a velocity at which a molecule can cross from condensed to dilute phase. As explained in the Results section, the chemical exchange rate from condensed to dilute phase k_{cd} will depend on the droplet size, namely on surface-to-volume ratio. A molecule in smaller droplet has a bigger chance to come to the actual interface and exchange phase, so k_{cd} is a rate at which molecules exchange phase.

FRAP experiment on the other hand depends a lot on the ratio of the bleaching area to the droplet size (10.1016/j.cell.2018.12.035 and several other papers cited in our manuscript). Indeed when only a part of the droplet is bleached, one probes the diffusion coefficient within condensed phase (effective diffusion coefficient sped up by inter-phase exchange). However, when full droplet is bleached, then in order to get recovery of the fluorescence one relies on chemical exchange from the dilute phase. This again highlights the significance of studying smaller droplets. This is discussed in Results for full-length FUS sample. We agree, however, as the condensate dynamics is mostly diffusion limited, that the FRAP recovery would still report on the condensed phase diffusion and not only on the exchange.

Having said this, it is indeed hard to compare REDIFINE k_{cd} and FRAP recovery side by side. This is why we tried to define in REDIFINE a global chemical exchange that takes into account the actual populations of protein and represents the normalized flux including the number of exchanged molecules between phases and not only rate. Accounting for surface-to-volume ratio, this gets somewhat more comparable with FRAP.

We have stressed this better in the revised manuscript.

6. The ability to measure protein concentrations (and partitioning coefficient) in a label-free manner could be quite valuable. It would be helpful to extend this measurement beyond FL FUS condensates.

We agree with this. NMR is one of the very few techniques that can extract the information on both water and biomolecules in condensates label-free. As in the current manuscript we introduce already multiple concepts, we wanted just to show an example for water among other applications. We think that this result is very exciting and deserves a separate study addressing partitioning coefficients and concentrations in various condensates under different conditions using the REDIFINE methodology.

7. The results in Fig 4 are informative, how do the quantified biophysical properties relate to factors such as molecule size, charge, or other characteristics?

As indicated in Figure 4, we find an interesting correlation between chemical exchange and the ratio between secondary structure prediction factor SSP (how structured protein is) and its charge. We also see a correlation between exchange and the charge of the protein which all makes sense given that LLPS is usually largely driven by electrostatic interactions. We tried to find some correlation with the molecule size, however this is not straightforward given that

diffusion in the condensed phase will depend not only on MW but on the protein shape and viscosity in the condensed phase.

8. The last section on soluble complexes raises the concern about how to distinguish whether the sample being measured contain condensates or complexes/oligomers in a REDIFINE experiment.

This is a good point and thank you for raising this concern. In the case of interaction, we have a biomolecule interchanging between bound and free form rather than physically restricted diffusion. When we tried to subject the data set acquired on soluble PTBP1:3xUCUCU complex (data from Figure 5b in the main text) to a REDIFINE model with restricted diffusion, the model completely failed to fit the data (Fig. R2). Also, one can see that the features of the curves present in Fig. 5b of the main text are different compared to a biphasic sample. Moreover, one can easily get insights from the turbidity/microscopy measurements to further alleviate discrepancies.

Fig. R2. REDIFINE data acquired on PTBP1:3xUCUCU = 1:1 soluble complex and ensuing fit to a REDIFINE model with restricted diffusion. The model including restricted diffusion failed to explain the data acquired on soluble protein:RNA complex. However, the diffusion data fit extremely well to a biexponential diffusion model with exchange (Fig. 5b in the main text).

Minor comments:

1. Please define NTD, FEXSY.

Thank you for noticing this. We have now defined these abbreviations.

2. Please be more specific on the number of fitting parameters, fitting error, and effect of heterogeneity in condensate size on the fitting results.

This is now further clarified in the revised manuscript. Fitting error is discussed in detail in the Methods section.

3. Please provide more information on factors that affect the Kurtosis effect and their significance in the context of the current findings.

The kurtosis effect is a mathematical term, a statistical metric quantifying the shape of a probability distribution. In the context of diffusion measurements, it represents the deviation from simple, Gaussian diffusion. As explained in detail in the SI, this effectively means that we don't have a monoexponential decay, but rather a mixture of multiple exponents. The actual kurtosis factor only qualitatively implies that there is a complex dynamics, and it would be affected by all the parameters that REDIFINE actually probes. Hence it is not very useful in the context of quantification of LLPS. Where it was largely applied was in the MRI imaging where $K > 0$ implied restricted motion of the water in the tissue and provided a contrast for acquiring images.

We discuss Kurtosis for historical reasons, relating our method to this. A detailed description is provided in the SI.

Reviewer 3

We would like to conclude by thanking again all the referees for their valuable and constructive comments. We believe that our clarifications and additions improved the quality and clarity of the manuscript, and that our revised document is now suitable for publishing in Nature Communications.

Sincerely and on behalf of all authors,

Mihajlo Novakovic, Leonidas Emmanouilidis and Frédéric Allain

11/02/2025

Institute of Biochemistry
Prof. Frédéric Allain

E-mail allain@bc.biol.ethz.ch
Address Hönggerbergring 64
8093 Zürich
Switzerland

Manuscript NCOMMS-24-59425-T

LLPS REDIFINE allows the biophysical characterization of multicomponent condensates without tags or labels

Mihajlo Novakovic^{1,*}, Nina Han^{1,2}, Nina C. Kathe^{1,2}, Yinan Ni^{1,2}, Leonidas Emmanouilidis^{1,*}, Frédéric H.-T. Allain^{1,*}

¹Department of Biology, Institute of Biochemistry, ETH Zurich, Zurich, Switzerland

²These authors contributed equally

*Corresponding authors. Email: mihajlo.novakovic@bc.biol.ethz.ch,
leonidas@bc.biol.ethz.ch, allain@bc.biol.ethz.ch

Response letter

Dear Dr. Cerullo,

Thank you for your email of Januar 28, 2025 containing the additional comments for the revision of our manuscript “**LLPS REDIFINE allows the biophysical characterization of multicomponent condensates without tags or labels**”. We went another step forward in addressing the Reviewer 2 concerns. We provided our responses in blue font.

Reviewer 1

As stated in my previous evaluation of this manuscript the presented work is interesting, important and very nicely executed and presented.

After seeing that the authors have addressed all major issues raised by me and the other reviewer I am in the opinion that the paper can be accepted for publication in Nature Communications.

We thank the reviewer for their very positive opinion of the manuscript. We did everything to address the comments of all reviewers and we are happy to see that our responses to both reviewers' points are acknowledged here.

Editorial notes according to Reviewer 2 comments

We had additional correspondence with Reviewer 2, and while some of their technical concerns have been addressed, they still have concerns regarding your manuscript. They find their main concerns about the applicability and biological relevance of REDIFINE are still unaddressed. They are of the opinion that studying condensates in an agarose gel is not a convincing context to make claims of biological relevance, and that there are marginal increases in unique information that REDIFINE provides compared to established assays. They consider the ability to measure protein/water concentrations in condensates to be the main strength of the technique, but would require further demonstration of this strength by applying your method to additional protein systems.

Given this feedback, editorially we find it important that you:

1. Justify the biological relevance.

The crucial step in the process of investigating biological condensation is to reconstitute a simplified assembly of the critical components *in vitro* and to test whether the assembly forms via LLPS. This is why a significant part of the literature represents studies that examine LLPS *in vitro*. Although there is increased development of the methodology to study this fascinating phenomenon, there is still a lack of approaches to study the dense milieu present in the condensed phase non-distractively and without using a label. Our manuscript introduces the REDIFINE approach to study biomolecular condensates non-destructively and without any label required. Although demonstrated with 5 different protein systems *in vitro*, our methodology has a huge potential *in cell*, which we started exploring.

We do understand that Referee 2 is primarily criticizing the biological relevance of the use of agarose as a stabilizing medium for the droplets. While *in vitro* studies generally consider LLPS systems in simple buffers, very often additional confinement and crowding agents are added to mimic complex cellular environments. Our choice to use agarose gel is discussed in detail in the publications previously published (<https://www.nature.com/articles/s41589-021-00752-3>, <https://www.nature.com/articles/s41589-024-01573-w>) and no such concerns were raised at that occasion. The same principles of using hydrogel were discussed also in <https://doi.org/10.1002/anie.201907278>. As the two phases differ in density, by definition condensates cannot exist in free aqueous environment long enough. Instead, they sediment at the bottom of the sample. This eventually leads to not only physical interaction with the glass altering the dynamics but also to the formation of significantly larger condensates. Specifically, in the case of FUS, due to the hydrophobic nature FUS droplets eventually disappear within minutes forming a thin layer of continuous condensed phase on the glass. One continuous condensed phase has significantly less surface-to-volume ratio than the same volume scattered over a number of smaller droplets affecting the chemical exchange between the phases. Therefore, as the exchange occurring on the surface of droplets is critical for the biological function of most condensates, it is important to study condensates in their droplet form rather than a single and large condensed phase.

0.5% agarose gel is so coarse that it barely affects the diffusion of small molecules such as water (we measure 2.265×10^{-9} m²/s in agarose vs literature values of 2.299×10^{-9} m²/s of bare water at 298K; <https://pubs.acs.org/doi/abs/10.1021/j100624a025>). Furthermore, we performed an extensive analysis to show that agarose does not affect the structure and behavior of the biomolecules that we are studying. This is summarized in Figure R1, where we show in panel a that the NMR spectra, depicting the structure of Nucleocapsid protein (Nu) in phosphate buffer without and with 0.5% agarose, are virtually identical. This confirms that Nucleocapsid, representing a highly complex protein consisting of structured and disordered domains (92 kDa), is not interacting with agarose at all. Figure R1b shows that the diffusion coefficients measured for the Nucleocapsid protein without and with agarose are also very simi-

lar (ca. 10% difference). As no complex diffusion dynamics is detected even using longer diffusion times (200 ms) as shown in Figure R1c, we can also confirm that Nu protein is not interacting with agarose. **We see therefore no reason to consider that the presence of agarose reduces the biological significance of our investigation.** The key role of agarose in our experiments is that it mimics a cytoskeleton, preventing droplets from merging and sedimenting. It therefore allows us to study condensates for a longer time in physiological buffers, and

Figure R1. Agarose is not affecting protein biomolecules. **a** 1D NMR spectra comparing Nucleocapsid protein (Nu) in absence and presence of 0.5% agarose gel. The nearly identical spectra imply that Nu is not interacting with agarose. **b,c** Diffusion of Nu protein in presence and absence of agarose at two diffusion times (75 ms and 200 ms). These plots show that Nu protein diffuses 10-15% slower in agarose compared to the buffer solution.

in essence, it represents a better system to depict what is happening in the cell. This is further

discussed in the Discussion section and shown in detail in Section 6 of Supplementary Information.

2. Show clear evidence of novelty/advance over established approaches.

We want to point out again that the most popular and widely used methods to study condensates require a tag or label ([https://www.cell.com/trends/biochemical-sciences/fulltext/S0968-0004\(20\)30146-8](https://www.cell.com/trends/biochemical-sciences/fulltext/S0968-0004(20)30146-8), <https://www.sciencedirect.com/science/article/pii/S0022283618307563>) that acts as a detection probe for a given technique. However, there are multiple reports illustrating that detection tags and labels influence both the morphology and dynamics of biomolecular condensates. This is written in the main text: “*Despite the evidence of the functional importance of phase-separation for these proteins, a wide spectrum of questions about the biophysical properties of these biomolecular condensates and their relation to (patho)physiology in biological systems remain unanswered,^{24,25} due to a lack of suitable experimental procedures that are not invasive and that can probe the biomolecules in their relevant form.^{26–28} Although multicolor labeling can provide a plethora of information about the condensates, the fluorescent tags often affect protein conformation and dynamics which can influence their phase separation introducing potential bias in the conclusions.^{29–35}”* with 7 references illustrating the potential problem of labels. Super-resolution microscopy is another emerging technique that can give unprecedented details of the droplet morphology, even the surface of the droplet beyond the diffraction limit, however, it also relies on fluorescent labels. Lately, electron microscopy has also gained a lot of momentum in studying condensates, yet it is destructive, involves freezing, and requires very involved experimental procedures. Studying droplet dynamics and diffusion using standard micro-rheology also requires fluorescent labels that can affect the behavior of biomolecules of interest. The content of condensates can be probed with MS techniques (we have also introduced LLPS CLIR-MS: <https://www.science.org/doi/full/10.1126/sciadv.adm7435>), and although very rewarding, MS techniques are also destructive and insensitive.

It was shown in several pioneering publications that NMR can provide unique structural insights (https://pubs.acs.org/doi/epdf/10.1021/jacs.9b12208?ref=article_openPDF, <https://www.pnas.org/doi/full/10.1073/pnas.2104897118>, [https://www.cell.com/molecular-cell/fulltext/S1097-2765\(15\)00702-9](https://www.cell.com/molecular-cell/fulltext/S1097-2765(15)00702-9), <https://www.nature.com/articles/s41594-019-0250-x>) and intrinsically does not require labels, setting it as an ideal technique to probe dynamic biomolecules in condensates. As shown in these papers, the NMR findings provided essential and exclusive information for the first time indicating the atomic structure of the proteins inside a condensed phase. Unfortunately, not all proteins can be produced in quantities to fill an NMR tube (circa 160 μ L) with pure condensed phase. We estimated that our method using agarose requires 50 to 100 times less material to obtain NMR spectra compared to this early pioneering approach. There are other emerging label-free techniques such as Atomic Force Microscopy, Optical Tweezers, Microfluidics, Mass Photometry that all come with their strengths and weaknesses. For example, AFM is limited only to the surface of condensates; optical tweezers are invasive, mass photometry is very unreliable and hard to interpret... In conclusion, no technique does it all and we need a diverse spectrum of methods to study these fascinating macromolecular structures. The **REDIFINE approach provides diffusion coefficients in intact dilute and condensed phases co-existing in equilibrium, partition coefficients, average droplet size, surface permeability and chemical exchange without any label required. Providing information about molecules inside and outside the condensed phase as well as the interface permeability, REDIFINE delivers unique information about the condensates. Multiple techniques combined and several sample preparations are needed to acquire the equivalent information. What is more, in addition to analyzing biomolecules, the REDIFINE approach can probe other condensate constituents such as water, buffer, and metabolites, within the same sample, all of which are not or**

hardly accessible to other techniques. Furthermore, REDIFINE can provide information about the binding dissociation constants of protein-RNA complexes, the binding kinetics in the dilute phase, and potentially even in the condensed phase.

We discuss the novelty and advances over current methodology throughout the manuscript, especially throughout the main text, in the Discussion section, and also in Supplementary Information. This is now further emphasized.

3. Demonstrate REDEFINE's applicability by applying their method to other protein systems and/or use your approach to provide new insights on understudied/challenging proteins.

We demonstrated the applicability of REDIFINE with five different proteins and around twenty different condensate systems (using different RNA molecules to induce phase separation, or different protein concentrations). We find this more than sufficient to validate and benchmark REDIFINE wide applicability. We also applied microscopy measurements to complement and compare with REDIFINE. Regarding the protein concentration by analyzing both biomolecules and water, we illustrated the applicability with one protein system – we dedicated a couple of paragraphs on this topic in the section “*LLPS REDIFINE applied to full-length FUS and water*”. As in the current manuscript, we introduce already multiple concepts, we wanted just to show an example of water integration among other applications. We think that this result is very exciting and deserves a separate study addressing partitioning coefficients and concentrations in various condensates under different conditions and exploring its applicability and limitations in detail.

Nevertheless, we have now measured structured water in another sample, formed by truncated FUS NTD protein. This is summarized in the Figure R2a. As in the example shown in the manuscript in Fig. 2 and Extended Data Fig. 3 we could again fit the data acquired on water to REDIFINE model and extract the dynamic information on structured water. Combining with protein fit shown in Supplementary Fig. 4, we could calculate an average concentration of FUS NTD in droplets to be around 5.3 mM for this particular sample of truncated FUS construct. Figure R2b compares water-detected data for FUS FL with truncated FUS NTD.

Figure R2. REDIFINE on water signal – application on FUS NTD. **a** Excellent fit of water detected data to REDIFINE model and ensuing fitting parameters **b** Comparison of water REDIFINE acquired on FL FUS and truncated FUS NTD illustrating the similarities and differences between the two condensed samples. As the error for D_{cond} is large, we will not further interpret the differences.

While water reports nearly identical average droplet size for two condensates and has identical diffusion in dilute phase, we can see that there is significantly less water in FUS NTD droplets (~16% less), which is expected as FUS FL has more polar pockets where water can be bound. Interacting less with FUS NTD, water exchanges somewhat faster with the bulk water compared to water in FL FUS.

This figure is now included as Supplementary Figure 5.

4. Properly acknowledge the limitations of the technology.

In our opinion, we discuss to quite an extent the limitations and applicability of our methodology. A couple of paragraphs is dedicated to this in Discussion section: “*LLPS REDIFINE requires for NMR analysis only one sample of unlabeled biomolecules stabilized in agarose gel (100-150 μ M in 130 μ l), although even lower concentrations can be used since NMR experiments with more extensive signal-averaging can be acquired. As the spectral resolution is not essential, LLPS REDIFINE is in theory applicable also at low-field benchtop machines with gradient capabilities.*

Currently, LLPS REDIFINE is based on a model that considers droplets of uniform size. Consequently, it provides information about the average droplet radius that best fits the experimental data. Although agarose-mimicking cytoskeleton prevents droplet fusion, in general, the droplets in the biphasic sample are somewhat heterogeneous in size. Although not encountered in this study, very heterogeneous size distributions might affect the fitting. Principally, if the distribution shape is assumed (normal or log-normal) LLPS REDIFINE could provide also a standard deviation of the distribution besides the average radius⁵⁶⁻⁵⁹ and this is currently being investigated. It should be mentioned that very fast exchange could also constrain the applicability of REDIFINE which is limited to the kinetic processes occurring on millisecond to second time scale. Furthermore, while the model accounts for the surface-to-volume ratio of the droplets, it is also possible to include different liquid droplet shapes, other than spheres. This could be ellipsoids and cylinders for example,³⁹ which could be applicable in the case of aggregated condensates and also for in cell (bacterial and eukaryotic) studies which are currently being explored.”

Furthermore, we are going in more depth in the section 8 “**Scope of LLPS REDIFINE**” in SI that goes in great details about the applicability and limitations. Not to mention that we discuss the experimental details and agarose gel preparation in the Methods section in the main text as well as in the sections 3 and 4 in SI.

We now emphasize further the requirements for sample preparation as a potential limitation in the Discussion section

We would like to conclude by thanking again all the referees and editors of Nature Communications. We did our best to address the critics and answer all editorial points, and we believe that our clarifications and additions render our manuscript suitable for publishing in Nature Communications.

Sincerely and on behalf of all authors,

Mihajlo Novakovic, Leonidas Emmanouilidis and Frédéric Allain